

# Universal tradeoff relation between speed, uncertainty and dissipation in nonequilibrium stationary states

**Izaak Neri**

Department of Mathematics, King's College London, Strand, London, WC2R 2LS, UK

## Abstract

We derive universal thermodynamic inequalities that bound from below the moments of first-passage times of stochastic currents in nonequilibrium stationary states of Markov jump processes in the limit where the two thresholds that define the first-passage problem are large. These inequalities describe a tradeoff between speed, uncertainty, and dissipation in nonequilibrium processes, which are quantified, respectively, with the moments of the first-passage times of stochastic currents, the splitting probability of the first-passage problem, and the mean entropy production rate. Near equilibrium, the inequalities imply that mean first-passage times are lower bounded by the Van't Hoff-Arrhenius law, whereas far from thermal equilibrium the bounds describe a universal speed limit for rate processes. When the current is proportional to the stochastic entropy production, then the bounds are equalities, a remarkable property that follows from the fact that the exponentiated negative entropy production is a martingale.

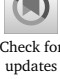

# 1 Introduction

In thermal equilibrium transitions between metastable states are activated by thermal fluctuations. The equilibrium transition rates satisfy the Van't Hoff-Arrhenius law [1,2]

$$k = \frac{1}{\langle T \rangle} = \nu e^{-\frac{E_{\mathrm{b}}}{\mathsf{T}_{\mathrm{env}}}}, \tag{1}$$

where the rate $k$ is the inverse of the mean first-passage time $\langle T \rangle$, $E_{\mathrm{b}}$ is the energy barrier that separates the two metastable states, $\mathsf{T}_{\mathrm{env}}$ is the temperature of the environment, and $\nu$ is a prefactor that has been determined, among others, by Kramers [1,3].

To speed up a process, an external agent can drive a system out of equilibrium. For example, in Fig. 1 we illustrate how external driving can increase the reaction rate in a nonequilibrium version of Kramers' model [3]. Other examples are the reduced travel times of self-propelled particles [4–8], the activated escape of a particle from a metastable state [9], enhanced relaxation rates in biomolecular diffusion processes [10], and enhanced reaction rates in nonequilibrium chemical reactions [11–14]. Since dissipation can increase the rate of a process, one may wonder whether there exists a generic speed limit on processes that are driven away from thermal equilibrium.

In the present paper, building on Ref. [15], we show that rate processes are governed by a universal tradeoff between dissipation, speed, and uncertainty. We quantify this tradeoff with generic inequalities on the moments of the first-passage times of stochastic currents with two thresholds. The derived inequalities are reminiscent of the thermodynamic uncertainty relations for first-passage times [16], but there exist also a couple of important distinctions. First, the trade-off relations derived in this paper quantify the uncertainty in the outcome of the process with the splitting probability of the first-passage problem, whereas the thermodynamic uncertainty relation quantifies uncertainty with the variance of the first-passage time. Second, the derived bounds are equalities when the current is the stochastic entropy production, and hence the derived first-passage inequalities are optimal in this case.

The paper is organised as follows: in Sec. 2, we state the main results of this paper. In Sec. 3, we discuss the setup for which the main results are derived, viz., stochastic currents in Markov jump processes. In Sec. 4, we derive the main results, within the setup of Markov jump processes, by using recent results on large deviations and martingales in stochastic thermodynamics. In Sec. 5 we provide an alternative derivation that is based on the theory of sequential hypothesis testing and which provides insights on extensions of the main results beyond Markov jump processes. In the following two Secs. 6 and 7, we relate the main results of this paper to results previously published in the literature and to the Van't Hoff-Arrhenius law, respectively. In Sec. 8, we illustrate with an example the tightness of the first-passage time bounds when the stochastic current is proportional to the stochastic entropy production. The paper ends with a discussion in Sec. 9 and after the discussion there are several appendices that contain technical details on the mathematical derivations.

# 2 Main results

The paper contains two main results. The first main result is an inequality that holds for the first-passage times of stochastic currents in stationary Markov jump processes. The second main result is an equality that holds for the first-passage times of stochastic currents that are proportional to the stochastic entropy production.

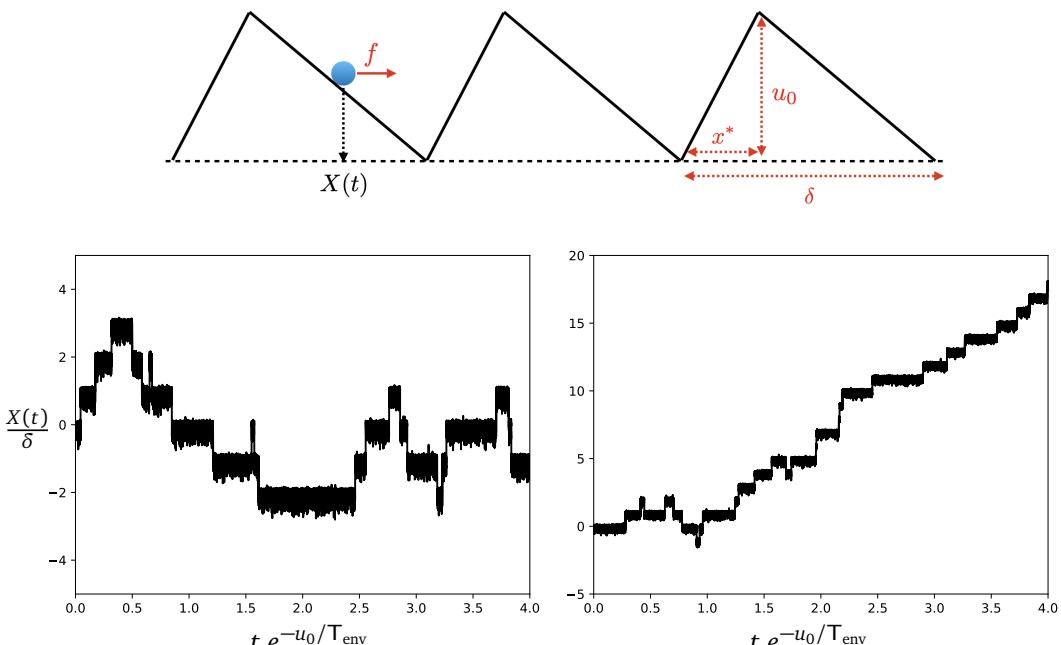

Figure 1: *Nonequilibrium version of Kramers' model exhibiting an increased reaction rate due to nonequilibrium driving*. Trajectories shown are for a reaction coordinate $X$ that solves the Langevin equation $\partial_t X(t) = (f - \partial_x u(X(t)))/\gamma + \sqrt{2T_{\text{env}}/\gamma}\,\xi(t)$, where $\xi(t) = dW(t)/dt$ is a delta-correlated white Gaussian noise term, and where $u(x)$ is a triangular potential with period $\delta$, i.e. $u(x) = u(\pm\delta)$, $u(x) = u_0 x/x^*$ if $x \in [0, x^*]$, and $u(x) = u_0(\delta - x)/(\delta - x^*)$ if $x \in [x^*, \delta]$. Left: equilibrium trajectory with $f = 0$. Right: nonequilibrium trajectory with $f\delta/T_{\text{env}} = 1$. The remaining parameters are set to $\delta = 5$, $\gamma = 1$, $x^* = 1$, $u_0 = 10$, and $T_{\text{env}} = 1$.

## 2.1 Bounds on the moments of first-passage times of stochastic currents

Let $J(t)$ be a stochastic current in a nonequilibrium, stationary process $X(t)$ and let

$$T_J = \inf\{t > 0 : J(t) \notin (-\ell_-, \ell_+)\} \tag{2}$$

be the first time when $J(t)$ leaves the open interval $(-\ell_-, \ell_+)$, where $t \geq 0$ is an index that labels the time and where $\ell_-, \ell_+ > 0$ are the threshold values of the first-passage problem.

In this paper we show that in the limit of large thresholds $-\ell_-$ and $\ell_+$ it holds that

$$\langle T_J^n \rangle \geq \left(\frac{\ell_+}{\ell_-}\frac{|\log p_-|}{\dot{s}}\right)^n (1 + o_{\ell_{\min}}(1)), \tag{3}$$

where

$$p_- = \mathbb{P}[J(T_J) \leq -\ell_-] \tag{4}$$

denotes the probability that the current $J$ goes below the negative threshold $-\ell_-$ before exceeding for the first time the positive threshold $\ell_+$, where $\dot{s}$ is the entropy production rate, and where $n \in \mathbb{N}$. The quantity $p_-$ is called the splitting probability. The averages $\langle \cdot \rangle$ are taken over repeated realisations of the stationary process $X$. We have used the little-o-notation $o_{\ell_{\min}}(1)$ to denote a function that converges to zero when $\ell_{\min} = \min\{\ell_-, \ell_+\} \to \infty$ while the ratio $\ell_-/\ell_+$ is kept fixed. Since we keep the ratio $\ell_-/\ell_+$ fixed, it holds that $o_{\ell_{\min}}(1) = o_{\ell_-}(1) = o_{\ell_+}(1)$. Equation (3) holds for $\langle J(t) \rangle > 0$; if $\langle J(t) \rangle < 0$, then $p_-$ should be replaced by $p_+ = \mathbb{P}[J(T_J) \geq \ell_+]$, $\ell_-$ with $\ell_+$, and vice versa.

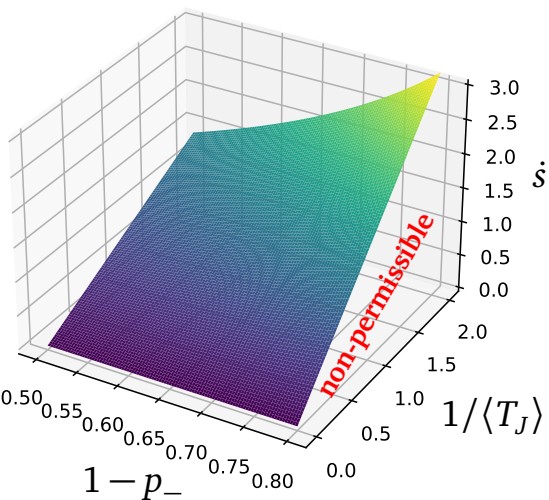

Figure 2: *Universal tradeoff between speed, uncertainty, and dissipation in nonequilibrium processes.* The three axes represent the speed ($1/\langle T_J \rangle$), uncertainty ($1 - p_-$), and dissipation ($\dot{s}$) in a nonequilibrium process $X$. The plotted surface is $\dot{s} = |\log p_-|/\langle T_J \rangle$; in the present example the thresholds are symmetric, $\ell_- = \ell_+$. Processes that are situated below the surface are physically nonpermissible as they violate the bound Eq. (3).

The inequality Eq. (3) describes a tradeoff between dissipation $\dot{s}$, speed $\langle T^n \rangle$, and the uncertainty in the outcome of the process that is quantified by $p_-$. It states that processes that are fast, precise, and have a small entropy production rate are physically not permissible. In Fig. 2 we illustrate this trade-off relation graphically by plotting a surface in a three-dimensional space delimiting the parameter regime that is physically not permissible.

Near equilibrium $\dot{s} \sim e^{-\frac{E_b}{T_{\text{env}}}}$ and $p_- \approx \ell_-/(\ell_+ + \ell_-)$. Consequently, Eq. (3) implies that $\langle T_J \rangle$ is lower bounded by the Van't Hoff-Arrhenius law, i.e.,

$$\langle T_J \rangle \geq \frac{1}{\nu} e^{\frac{E_b}{T_{\text{env}}}} . \tag{5}$$

On the other hand, far from thermal equilibrium the right hand side of Eq. (3) goes below $\frac{1}{\nu} e^{\frac{E_b}{T_{\text{env}}}}$ implying that dissipation can increase the reaction rate $k = 1/\langle T_J \rangle$, as we illustrate in Fig. 1 for a nonequilibrium version of Kramer's model [3].

Taken together, the Eq. (3) states that we can speed up a process by driving it out of equilibrium, but there exists a universal speed limit that is determined by the rate of dissipation and the amount of fluctuations in the process.

## 2.2 Equality for the moments of first-passage times of entropy production

If $J(t) = S(t)$ with $S(t)$ the stochastic entropy production [17–19], then the equality sign in Eq. (3) holds, viz.,

$$\langle T_S^n \rangle = \left( \frac{\ell_+}{\ell_-} \frac{|\log p_-|}{\dot{s}} \right)^n (1 + o_{\ell_{\min}}(1)) . \tag{6}$$

This remarkable property follows form the fact that $e^{-S(t)}$ is a martingale [20–23], which implies the formula $p_- = e^{-\ell_-}(1 + o_{\ell_{\min}}(1))$ [21, 23].

Note that the definition (2) together (6) implies that the equality sign in Eq. (3) also holds when $J(t) = cS(t)$, with $c$ a constant that is independent of $\ell_-$ and $\ell_+$.

The Eq. (6) implies that the bound Eq. (3) is tight when the stochastic current is proportional to the stochastic entropy production ($J = cS$), and this is one of the main advantages of the bound (3) with respect to other tradeoff inequalities reported in the literature, such as, the thermodynamic uncertainty relation for first-passage times that quantifies uncertainty in terms of the variance of the first-passage time [16].

# 3 System setup

We consider a stationary Markov jump process $X(t)$ defined on a discrete set $\mathcal{X} \ni X(t)$ and in continuous time $t \geq 0$. The dynamics of $X(t)$ consists of a sequence of jumps with rates that are determined by a Markov transition rate matrix $w_{x \to y}$ with $x, y \in \mathcal{X}$ [24]. We assume that $X(t)$ has a unique stationary probability distribution $p_{\mathrm{ss}}(x)$ that satisfies $p_{\mathrm{ss}}(x) > 0$ for all $x \in \mathcal{X}$, and we assume that the process is reversible in the sense that $w_{x \to y} > 0$ if and only if $w_{y \to x} > 0$.

Stochastic currents $J(t) = J(X_0^t)$ are real-valued functionals defined on the set of trajectories $X_0^t$ with the following two properties:

(i) $J$ is time extensive, i.e.,

$$\langle J(t) \rangle = \bar{j}\, t \,, \tag{7}$$

where $\bar{j}$ is a nonzero current rate. Without loss of generality we can assume that $\bar{j} > 0$.

(ii) $J$ is odd under time-reversal, i.e.,

$$J(\Theta_t(X_0^t)) = -J(X_0^t)\,, \tag{8}$$

where the time-reversal operation $\Theta_t$ maps trajectories $X_0^t$ on their time-reversed trajectory $(X^\dagger)_0^t$ with entries $X^\dagger(\tau) = X(t - \tau)$. Note that this implies $J(0) = 0$.

In a Markov jump process, stochastic currents take the form

$$J(t) = \sum_{x,y \in \mathcal{X}} c_{x,y} J_{x \to y}(t), \tag{9}$$

with coefficients $c_{x,y} \in \mathbb{R}$ and with $c_{x,x} = 0$. The edge currents

$$J_{x \to y}(t) = N_{x \to y}(t) - N_{y \to x}(t) \tag{10}$$

denote the difference between the number of times $N_{x \to y}(t)$ the process has jumped from the $x$-th state to the $y$-th state in the trajectory $X_0^t$ and the number of reverse jumps $N_{y \to x}(t)$ from the $y$-th to the $x$-th state in the same trajectory.

The stochastic entropy production $S$ is defined by the ratio [17, 19]

$$S(t) = \log \frac{p(X_0^t)}{p(\Theta_t(X_0^t))} \tag{11}$$

between the probability distributions of the trajectory $X_0^t$ in the forward and backward dynamics, better known as the Radon-Nikodym derivative [21, 25, 26]. For a stationary process, the index $t$ in the map $\Theta_t$ of Eq. (11) is immaterial, and we can replace $\Theta_t$ by $\Theta$. Notice that we

use natural units for which the Boltzmann constant is set equal to one. It is possible to write the stochastic entropy production in the form Eq. (9), viz.,

$$S(t) = \frac{1}{2} \sum_{x,y \in \mathcal{X}} \log \frac{p_{\text{ss}}(x) w_{x \to y}}{p_{\text{ss}}(y) w_{y \to x}} J_{x \to y}(t) , \tag{12}$$

where $p_{\text{ss}}(x)$ is the probability distribution of $X(t)$ in the stationary state. In the definition of the entropy production we require that the process is reversible, i.e., if $w_{x \to y} > 0$ then also $w_{y \to x} > 0$. A useful property that we will use repeatedly is that the exponentiated negative entropy production $e^{-S(t)}$ is a martingale, see [20–23].

Since the process is stationary, the entropy production rate $\dot{s}$ is given by

$$\langle S(t) \rangle = \dot{s}\, t . \tag{13}$$

For systems that are weakly coupled to an environment in thermal equilibrium, the entropy production rate equals the dissipation rate [17,19,27], which clarifies the physical significance of the process $S(t)$. In the literature, the latter property is often referred to as the principle of local detailed balance [28, 29].

## 4 First-passage time bounds from large deviation theory

We derive the main results of this paper, given by Eqs. (3) and (6), with large deviation theory.

Stochastic currents $J(t)$ in Markov jump processes satisfy a large deviation principle. This means that for large enough times $t$, the probability distribution of $J/t$ takes the form [30,31]

$$p_{J/t}(z) = e^{-t \mathcal{J}(z)(1 + o_t(1))} , \tag{14}$$

where $o_t(1)$ is a function that converges to zero when $t$ is large enough, and where $\mathcal{J}(z)$ is the large deviation function of the current. In Eq. (14), the normalisation constant is contained in the $o_t(1)$ term that appears in the argument of the exponential. The large deviation function $\mathcal{J}(z) \geq 0$ is a convex function that takes its minimum value when $J/t = \bar{j}$, i.e., $\mathcal{J}(\bar{j}) = 0$.

An immediate consequence of Eq. (14) is that

$$\langle T_J^n \rangle = \left( \frac{\ell_+}{\bar{j}} \right)^n (1 + o_{\ell_{\min}}(1)) . \tag{15}$$

Indeed, since $J(t)$ satisfies the large-deviation principle Eq. (14), $J(t)$ converges with probability one to $\bar{j}t$, viz.,

$$\frac{J(t)}{t} = \bar{j}(1 + o_t(1)) . \tag{16}$$

Consequently, the first-passage time given by Eq. (2) is deterministic for large values of $\ell_{\min}$, and since $\bar{j} > 0$ we obtain

$$T_J = \frac{\ell_+}{\bar{j}}(1 + o_{\ell_{\min}}(1)) , \tag{17}$$

which implies Eq. (15), as long as for finite threshold values $\ell_{\min}$ the distribution of $T_J$ has fast enough decaying tails.

To complete the derivation of the main results, we derive in Sec. 4.1 a lower bound for the splitting probability $p_-$, in particular, we show that

$$p_- \geq \exp\left( -\frac{\ell_- \dot{s}}{\bar{j}}(1 + o_{\ell_{\min}}(1)) \right) , \tag{18}$$

which together with (15) implies Eq. (3).

In Sec. 4.2 we show that for $J = S$ the inequality (18) becomes an equality, leading to (6).

## 4.1 Bound on the splitting probability $p_-$

We derive the bound Eq. (18) for the probability $p_-$ that $J$ hits the negative boundary first, which together with (15) readily implies the main result Eq. (3).

For stationary Markov jump processes, it was shown that $\mathcal{J}(z)$ is bounded from above by [32–34]

$$\mathcal{J}(z) \le \frac{\dot{s}}{4}(z/\bar{j}-1)^2 . \tag{19}$$

In what follows, we show that the inequality (18) follows from this fundamental bound.

The splitting probability $p_-$ can be expressed as follows,

$$
\begin{aligned}
p_- = \mathbb{P}[J(T_J) \le -\ell_-] = \mathbb{P}[J(t) \le -\ell_-] &- \mathbb{P}[J(t) \le -\ell_- \wedge J(T_J) \ge \ell_+] \\
&+ \mathbb{P}[J(t) \ge -\ell_- \wedge J(T_J) \le -\ell_-] ,
\end{aligned}
\tag{20}
$$

where $\wedge$ is a short hand notation for the logical conjunction. Since probabilities are positive, we obtain the bound

$$p_- \ge \mathbb{P}[J(t) \le -\ell_-] - \mathbb{P}[J(t) \le -\ell_- \wedge J(T_J) \ge \ell_+] . \tag{21}$$

Moreover, using that for large enough thresholds the probability that $J(t)$ goes below the threshold $\ell_-$ after it went above the threshold $\ell_+$ is vanishingly small, we obtain the inequality

$$p_- \ge \mathbb{P}[J(t) \le -\ell_-](1 + o_{\ell_{\min}}(1)) \tag{22}$$

that holds for all $t \ge 0$.

We can express the right-hand side of Eq. (22) in terms of $p_{J/t}(z)$, i.e.,

$$p_- \ge \int_{-\infty}^{-\ell_-/t} \mathrm{d}z\, p_{J/t}(z) = \int_{-\infty}^{-\ell_-/t} \mathrm{d}z\, e^{-t\mathcal{J}(z)(1+o_t(1))} . \tag{23}$$

Using the bound Eq. (19) in Eq. (23) and setting $\tau = t/\ell_-$, we obtain

$$p_- \ge \int_{-\infty}^{-1/\tau} \mathrm{d}z\, \exp\left(-\frac{1}{4}\ell_- \dot{s}\tau\left[\frac{z}{\bar{j}}-1\right]^2 (1 + o_{\ell_{\min}}(1))\right) , \tag{24}$$

where we have also interchanged $o_t(1)$ with $o_{\ell_{\min}}(1)$. This is possible since the results of this paper hold for $\ell_{\min} \to \infty$ while keeping the ratio $\ell_-/\ell_+$ fixed. In Eq. (24) this limit corresponds with $\ell_- \to \infty$ while keeping the ratio $t/\ell_- = \tau$ fixed. In this limit, it holds that $o_t(1) = o_{\ell_{\min}}(1)$, and therefore we can interchange these two symbols.

For large values of $\ell_-$, the expression Eq. (24) is a saddle point integral, and hence it is determined by the maximum of the exponent attained at $z = -1/\tau$, i.e.,

$$p_- \ge \exp\left(-\frac{1}{4}\ell_- \dot{s}\tau\left[\frac{1}{\tau\bar{j}}+1\right]^2 (1 + o_{\ell_{\min}}(1))\right) . \tag{25}$$

Since the above inequality holds for arbitrary $\tau$, we can take the maximum of the right-hand side,

$$p_- \ge \max_{\tau \ge 0} \exp\left(-\frac{1}{4}\ell_- \dot{s}\tau\left[\frac{1}{\tau\bar{j}}+1\right]^2 (1 + o_{\ell_{\min}}(1))\right) . \tag{26}$$

For $\tau \ge 0$, the minimum value of the function $\tau\left(\frac{1}{\tau\bar{j}}+1\right)^2$ is reached when $\tau = 1/\bar{j}$, leading to the bound Eq. (18) that we were meant to derive.

## 4.2 An equality for $p_-$ that follows from the martingality of $e^{-S}$

To derive Eq. (6), we show that when $J = S$, then the equality in Eq. (18) satisfied, i.e.,

$$p_- = e^{-\ell_-(1+o_{\ell_{\min}}(1))} \,. \tag{27}$$

Eq. (27) together with Eq. (15), readily implies Eq. (6).

The fact that $p_-$ is universal and only depends on the threshold $\ell_-$ is a remarkable fact that is a direct consequence of the martingale property of $e^{-S(t)}$ [20, 21, 23]. Indeed, since the process $e^{-S(t)}$ is a martingale and since $T_S$ is a first-passage time with two thresholds, the integral fluctuation relation at stopping times [23]

$$\langle e^{-S(T_S)} \rangle = 1 \tag{28}$$

applies, see Corollary 2 of the Appendix of Ref. [23]; a related, albeit not identical, relation was reported in [35]. Using that $\mathbb{P}[T_S < \infty] = 1$, the Eq. (28) also reads

$$p_- \langle e^{-S(T_S)} \rangle_- + p_+ \langle e^{-S(T_S)} \rangle_+ = 1 \,, \tag{29}$$

where $\langle \cdot \rangle_-$ and $\langle \cdot \rangle_+$ denote averages over those trajectories that terminate at the negative and positive threshold values, respectively. Using that for $\ell_-, \ell_+ \gg 1$ it holds that $S(T_S) = \ell_\pm(1 + o_{\ell_{\min}}(1))$, we obtain

$$p_- e^{\ell_-(1+o_{\ell_{\min}}(1))} + p_+ e^{-\ell_+(1+o_{\ell_{\min}}(1))} = 1 \,, \tag{30}$$

and for $\ell_+ \gg 1$ this simplifies into Eq. (27).

# 5 First-passage time bounds from the asymptotic optimality of sequential probability ratio tests

In the previous section, we have derived the main results Eqs. (3) and (6) within the setup of stationary Markov jump processes. In the present section, we derive the main results within the framework of sequential hypothesis testing. With sequential hypothesis testing theory, we can derive partial results in an extremely general setting. These partial results are interesting in their own right, and they also pave the way to derive Eqs. (3) and (6) in a setup more general than Markov jump processes.

## 5.1 Review of sequential hypothesis testing

As pointed out in Ref. [15], first-passage problems for stochastic currents with two thresholds are sequential hypothesis tests that decide on the arrow of time, and first-passage problems for entropy production are sequential probability ratio tests. Therefore, we can use the theory of sequential hypothesis testing to derive bounds on the moments of first-passage times of stochastic currents. We provide a brief review of the theory of sequential hypothesis testing, focusing on the asymptotic optimality of sequential probability ratio tests.

Sequential hypothesis tests are statistical hypothesis tests that take a decision $D$ about the true hypothesis $H$ at a random stopping time $T$. The general setup goes as follows [36, 37]. There is an observation process $X(t)$ whose statistics are determined by one of two possible probability measures $\mathbb{P}_+$ or $\mathbb{P}_-$ corresponding to two hypotheses $H = +$ and $H = -$, respectively. A sequential hypothesis test is a pair $(T, D)$, where $T$ is a stopping time relative to the process $X$, and $D \in \{-, +\}$ is a decision variable defined on the set of trajectories $X_0^T$ up to the decision time $T$. The error reliabilities of the test are

$$p_- = \mathbb{P}_+[D = -] \quad \text{and} \quad p_+^\dagger = \mathbb{P}_-[D = +] \,, \tag{31}$$

where $\mathbb{P}_+[D=-] = \mathbb{P}[D=-|H=+]$ and $\mathbb{P}_-[D=+] = \mathbb{P}[D=+|H=-]$.

Given certain maximally allowed error probabilities $\alpha_-$ and $\alpha_+$, we define the set

$$\mathcal{C}_{\alpha_-,\alpha_+} = \left\{ (T,D): p_- \le \alpha_+,\ p_+^\dagger \le \alpha_-,\quad \langle T|H=+\rangle < \infty,\ \langle T|H=-\rangle < \infty \right\} \tag{32}$$

of all sequential hypothesis tests that meet the required constraints on the error reliabilities and with finite expected decision times under both hypotheses. We say that a sequential hypothesis test is optimal if it is an element of $\mathcal{C}_{\alpha_-,\alpha_+}$ and it minimises the mean decision times $\langle T|H=+\rangle$ and $\langle T|H=-\rangle$.

For general observation processes $X(t)$, the optimal sequential hypothesis test is not known. However, in the asymptotic limit of small maximally allowed error probabilities $\alpha_-$ and $\alpha_+$ the optimal test is known and given by the sequential probability ratio test [37]. The sequential probability ratio test was first introduced by Wald for observation processes that are a sequence of independent and identically distributed random variables [38], and subsequently, Wald and Wolfowitz proved the optimality of the sequential probability ratio in the latter setup [39]. In a later work [40], Lai proved the asymptotic optimality of sequential probability ratio tests for general observation processes.

Let

$$\Lambda(t) = \log \frac{p^+(X_0^t)}{p^-(X_0^t)}, \tag{33}$$

be the log-likelihood ratio process, which should be understood as the logarithm of the Radon-Nikodym derivative of the probability measure $\mathbb{P}_+$ with respect to the probability measure $\mathbb{P}_-$, both constrained on the sub-$\sigma$-algebra generated by the trajectories $X_0^t$. Loosely said, $\Lambda(t)$ is the logarithm of the ratio of the probability densities $p^+(X_0^t)$ and $p^-(X_0^t)$ associated to the trajectories $X_0^t$, which clarifies the notation in Eq. (33). The sequential probability ratio test is then the first-passage problem $T_\Lambda$ (see Eq. (2) for the definition of first-passage times) with thresholds $-\ell_-$ and $\ell_+$ that determine the error probabilities $p_-$ and $p_+^\dagger$. When $\Lambda$ is a continuous process, then

$$\ell_- = \log[(1-p_+^\dagger)/p_-], \quad \ell_+ = \log[(1-p_-)/p_+^\dagger]. \tag{34}$$

We formulate a lemma and a theorem about the asymptotic properties of sequential hypothesis tests and the asymptotic optimality of sequential probability ratio tests. We first consider Lemma 3.4.1 in [37] that derives an asymptotic lower bound for the moments of the decision times of sequential hypothesis tests.

**Lemma 1** (Asymptotic lower bounds for the moments of decision times in sequential hypothesis tests). *Let $\delta = (T,D)$ be a sequential hypothesis test in the set $\mathcal{C}_{\alpha_-,\alpha_+}$. We assume that $\Lambda(t) \in \mathbb{R}$ and $1/\Lambda(t) \in \mathbb{R}$ for all $t \ge 0$. We assume that there exists a nonnegative increasing function $\psi(t)$ with $\psi(\infty) = \infty$ such that*

$$\lim_{t\to\infty} \frac{\Lambda(t)}{\psi(t)} = \overline{\lambda}_+, \quad (\mathbb{P}_+\text{-almost surely}); \quad \lim_{t\to\infty} \frac{\Lambda(t)}{\psi(t)} = -\overline{\lambda}_-, \quad (\mathbb{P}_-\text{-almost surely}), \tag{35}$$

*with $\overline{\lambda}_-, \overline{\lambda}_+ \in (0,\infty)$. Moreover, we assume that for all finite $\tau$*

$$\mathbb{P}_+\left[\sup_{t\in[0,\tau]}\Lambda(t) < \infty\right] = 1, \quad \mathbb{P}_-\left[-\inf_{t\in[0,\tau]}\Lambda(t) < \infty\right] = 1. \tag{36}$$

*Under these assumptions, it holds that for all $\epsilon > 0$*

$$\lim_{\alpha_{\max}\to 0} \inf_{\delta\in\mathcal{C}(\alpha_-,\alpha_+)}\mathbb{P}_+\left[T > (1-\epsilon)\Psi\left(|\log\alpha_-|/\overline{\lambda}_+\right)\right] = 1, \tag{37}$$

$$\lim_{\alpha_{\max}\to 0} \inf_{\delta\in\mathcal{C}(\alpha_-,\alpha_+)}\mathbb{P}_-\left[T > (1-\epsilon)\Psi\left(|\log\alpha_+|/\overline{\lambda}_-\right)\right] = 1, \tag{38}$$

*where $\Psi(t)$ is the inverse of $\psi(t)$, i.e., $\Psi(\psi(t)) = t$. Moreover, for all $n > 0$*

$$\lim_{\alpha_{\max} \to 0} \inf_{\delta \in \mathcal{C}(\alpha_-, \alpha_+)} \langle T^n | H = + \rangle \geq \left( \Psi \left( |\log \alpha_-| / \overline{\lambda}_+ \right) \right)^n (1 + o_{\alpha_{\max}}(1)), \tag{39}$$

$$\lim_{\alpha_{\max} \to 0} \inf_{\delta \in \mathcal{C}(\alpha_-, \alpha_+)} \langle T^n | H = - \rangle \geq \left( \Psi \left( |\log \alpha_+| / \overline{\lambda}_+ \right) \right)^n (1 + o_{\alpha_{\max}}(1)). \tag{40}$$

Second, we consider Theorem 3.4.2 in [37] for the asymptotic optimality of the sequential probability ratio test. Contrarily to Lemma 1, this theorem provides an equality for the moments of first-passage times and for this reason we will need to replace the almost sure convergence conditions Eqs. (35) by the stronger $r$-quick convergence condition. Let

$$L_\epsilon(Y(t)) = \sup \{ t > 0 : |Y(t)| > \epsilon \}, \tag{41}$$

be the last entry time of a real-valued stochastic process $Y(t) \in \mathbb{R}$ into an interval $[-\epsilon, \epsilon]$, with $\sup \{\phi\} = 0$. We say that $Y(t)$ converges $r$-quickly to 0 in $\mathbb{P}_+$ if $\langle L_\epsilon^r | H = + \rangle < \infty$ for every $\epsilon > 0$.

**Theorem 1** (Asymptotic optimality of sequential probability ratio tests). *We assume that*

$$\lim_{t \to \infty} \frac{\Lambda(t)}{\psi(t)} = \overline{\lambda}_+, \quad (r-\text{quickly in } \mathbb{P}_+); \quad \lim_{t \to \infty} \frac{\Lambda(t)}{\psi(t)} = -\overline{\lambda}_-, \quad (r-\text{quickly in } \mathbb{P}_-) \tag{42}$$

*where $r$ is a natural number. It holds then that*

- *for any finite threshold values $\ell_-$ and $\ell_+$,*

$$\langle T_\Lambda^r | H = \pm \rangle < \infty; \tag{43}$$

- *for all $m \in (0, r]$,*

$$\langle T_\Lambda^m | H = \pm \rangle = \left( \Psi \left( \ell_\pm / \overline{\lambda}_\pm \right) \right)^m \left( 1 + o_{\ell_{\min}}(1) \right); \tag{44}$$

- *if $\ell_- = |\log p_-|(1 + o_{\ell_{\min}}(1))$ and $\ell_+ = |\log p_+^\dagger|(1 + o_{\ell_{\min}}(1))$, then for all $m \in (0, r]$*

$$\langle T_\Lambda^m | H = + \rangle = \left( \Psi \left( |\log p_+^\dagger| / \overline{\lambda}_+ \right) \right)^m \left( 1 + o_{\ell_{\min}}(1) \right) \tag{45}$$

*and*

$$\langle T_\Lambda^m | H = - \rangle = \left( \Psi \left( |\log p_-| / \overline{\lambda}_- \right) \right)^m \left( 1 + o_{\ell_{\min}}(1) \right). \tag{46}$$

## 5.2 Derivation of the first-passage bound Eq. (3) based on Lemma 1

We use Lemma 1 to derive Eq. (3). However, as will become soon evident, Lemma 1 is not equivalent to Eq. (3), as to derive Eq. (3) we also need to relate $p_+^\dagger$ to $p_-$.

Let $\mathbb{P}$ denote the probability measure of events in the forward dynamics and let $\mathbb{P} \circ \Theta$ be the probability measure of events in the time-reversed dynamics. Setting $\mathbb{P}_+ = \mathbb{P}$, $\mathbb{P}_- = \mathbb{P} \circ \Theta$, and $\psi(t) = t$, we obtain according to definition (11) that $\Lambda(t) = S(t)$ and $\overline{\lambda}_+ = \dot{s}$. Since $J$ is a stochastic current it changes sign under time-reversal and therefore the pair $(T_J, D_J)$, with $T_J$ as defined in Eq. (2) and $D_J = \text{sign}(J(T_J))$, is a sequential hypothesis test corresponding to the two probability measures $\mathbb{P}$ and $\mathbb{P} \circ \Theta$ [15]. Replacing in Eq. (39) the $\alpha_-$ by $p_+^\dagger$ and the $o_{\alpha_{\max}}(1)$ by $o_{\ell_{\min}}(1)$, we obtain [15]

$$\langle T_J^n \rangle \geq \left( \frac{|\log p_+^\dagger|}{\dot{s}} \right)^n (1 + o_{\ell_{\min}}(1)). \tag{47}$$

In Appendix E, we derive using heuristic mathematical arguments the equality

$$\frac{\ell_-}{\ell_+} = \frac{|\log p_-|}{|\log p_+^\dagger|}(1 + o_{\ell_{\min}}(1)) , \tag{48}$$

for currents $J$ in stationary Markov jump processes $X$ taking values in a finite set $\mathcal{X}$. Multiplying the right-hand side of Eq. (47) with

$$1 = \left(\frac{\ell_+}{\ell_-}\frac{|\log p_-|}{|\log p_+^\dagger|}\right)^n , \tag{49}$$

we obtain Eq. (3), which concludes the derivation.

The partial result Eq. (47) is interesting in its own right as it is an extremely, general relation that has been derived with full mathematical rigour. Indeed, Lemma 1 holds for processes $X$ that are reversible, in the sense that the stochastic entropy production $S(t)$ is well defined, and obey a weak stationary condition, in the sense that $S(t)/t$ converges almost surely to a deterministic limit. Remarkably, we do not require a large deviation principle for $J$, and we do not even require a large deviation principle for $S$.

To obtain Eq. (3) from Eq. (47), we have used Eq. (48). Note that (48) has not been derived with the same mathematical rigour as (47), and it is not clear whether Eq. (48) is valid beyond the setup of stationary Markov jump processes. However, Eq. (3) can be interpreted as a tradeoff relation between dissipation, speed, and uncertainty, whereas the interpretation of Eq. (47) as a trade-off relation is less clear, as $p_+^\dagger$ is the splitting probability in the time-reversed process.

### 5.3 Derivation of the asymptotic equality Eq. (6) based on Theorem 1

We set again $\mathbb{P}_+ = \mathbb{P}$, $\mathbb{P}_- = \mathbb{P} \circ \Theta$, and $\psi(t) = t$, obtaining $\overline{\lambda}_+ = \dot{s}$ and $\Psi(t) = t$. Therefore, Eq. (45) reads

$$\langle T_S^n \rangle = \left(\frac{|\log p_+^\dagger|}{\dot{s}}\right)^n (1 + o_{\ell_{\min}}(1)). \tag{50}$$

In Sec. 4.2 we have shown that

$$\ell_- = |\log p_-|(1 + o_{\ell_{\min}}(1)), \tag{51}$$

which follows readily from the martingale property of $e^{-S(t)}$. Analogously, one can show that [21]

$$\ell_+ = |\log p_+^\dagger|(1 + o_{\ell_{\min}}(1)). \tag{52}$$

Multiplying the right-hand side of Eq. (50) with

$$1 = \left(\frac{\ell_+}{\ell_-}\frac{|\log p_-|}{|\log p_+^\dagger|}\right)^n , \tag{53}$$

we obtain Eq. (6), which completes the derivation. Note that because of the martingale property of $e^{-S}$ the Eq. (6) can be derived with full mathematical rigour in a very general setup.

## 6 Connections between Eq. (3) and other thermodynamic trade-off relations

We point out connections between Eq. (3) and thermodynamic trade-off inequalities that appeared before in the literature.

## 6.1 Decision making in the arrow of time

Equation (9) in Ref. [15] implies Eq. (47). Indeed, Equation (9) in Ref. [15] states that in the limit $p_+^\dagger \to 0$,

$$\dot{s} \geq \frac{(1-2p_+^\dagger)\log\big((1-p_+^\dagger)/p_+^\dagger\big)}{\langle T_J \rangle} . \tag{54}$$

which is equivalent to Eq. (47) in this limit.

The main distinction between the results in Ref. [15] and those of the present paper, is that the Eqs. (3) and (6) involve $p_-$, while the results of [15] involve $p_+^\dagger$. This distinction is relevant as $p_+^\dagger$ involves fluctuations of the process in a time-reversed dynamics that is not always accessible.

## 6.2 Dissipation-time uncertainty relation

Eq. (3) is related to the so-called dissipation-time uncertainty relation that states

$$\langle T_J \rangle \geq \frac{1}{\dot{s}} , \tag{55}$$

in the limit $|\log p_+^\dagger| \gg 1$ [41, 42].

The dissipation-time uncertainty relation is a loose bound when compared to the bounds Eqs. (3) and Eq. (47). Indeed, comparing Eq. (55) with (3), or better Eq. (55) with (47), we conclude that

$$\langle T_J \rangle \geq \frac{c}{\dot{s}}(1 + o_{\ell_{\min}}(1)) \tag{56}$$

holds for any prefactor $c \geq 0$. This is because the prefactor in Eq. (47) is $c = |\log p_+^\dagger|$ and thus diverges when $p_+^\dagger$ is small.

## 6.3 Thermodynamic uncertainty relations

The bound (3) follows from the bound Eq. (19) on the large deviation function of a stochastic current. Since also the thermodynamic uncertainty relations have been derived using the bound (19), see Refs. [32, 33, 43], we discuss here how the bound Eq. (3) is related to thermodynamic uncertainty relations.

The thermodynamic uncertainty relation bounds from below the Fano factor of stochastic currents, i.e., [33, 43]

$$\frac{\sigma_J^2}{2\bar{j}^2} \geq \frac{1}{\dot{s}} , \tag{57}$$

where $\bar{j}$ is the current rate and

$$\sigma_J^2 = \lim_{t \to \infty} \frac{1}{t}\big(\langle J^2(t) \rangle - \langle J(t) \rangle^2\big) . \tag{58}$$

A first-passage time thermodynamic uncertainty relation was derived in Ref. [16], viz.,

$$\frac{\langle T_J^2 \rangle - \langle T_J \rangle^2}{2\langle T_J \rangle} \geq \frac{1}{\dot{s}}(1 + o_{\ell_{\min}}(1)). \tag{59}$$

The bounds Eqs. (3), (57) and (59) all express a nonequilibrium tradeoff between dissipation, speed, and uncertainty. The differences between these bounds is in how they quantify speed and uncertainty. The thermodynamic uncertainty relation Eq. (57) quantifies speed with $\bar{j}$ and uncertainty with $\sigma_J^2$, the first-passage time uncertainty relation Eq. (59) quantifies speed

with $\langle T_J \rangle$ and uncertainty with $\langle T_J^2 \rangle - \langle T_J \rangle^2$, and the bound Eq. (3) quantifies speed with $\langle T_J \rangle$ and uncertainty with $p_-$.

An important distinction between the thermodynamic uncertainty relations, Eqs. (57) and Eq. (59), and the bound Eq. (3) on the moments of first-passage times, is that the latter is tight when $J = S$ while the former is loose. Indeed, if $J(t) = S(t)$, then Eq. (3) becomes the equality Eq. (6), whereas the Eqs. (57) and Eq. (59) are in general not equalities, even not when $J(t) = S(t)$ [22,44]. How is this possible, given that the relations (3), (57), and (59) are all derived from the same bound, viz., Eq. (19) on the large deviation function? We can understand this as follows. Eq. (3) is obtained from evaluating the bound (19) at the value $z = -\bar{j}$, while Eqs. (57) and (59) rely on the properties of the large deviation function in the vicinity of the point $z = \bar{j}$, in particular, the derivatives of the large deviation function at this point. As observed in Ref. [32], the large deviation function bound Eq. (19) is tight when $J = S$ and $z = -\dot{s}$, while this is not the case for the slope of the bound at $z = \dot{s}$, as the large deviation function of $S$ is in general not a parabola.

The tightness of the bound (19) for $J = S$ at $z = -\dot{s}$ can also be understood from the Gallavotti-Cohen fluctuation relation [45]

$$\mathcal{J}(z) - \mathcal{J}(-z) = -z. \tag{60}$$

For $z = -\dot{s}$, the Gallavotti-Cohen relation implies that $\mathcal{J}(-\dot{s}) = \dot{s}$ as $\mathcal{J}(\dot{s}) = 0$. One verifies readily that the right hand side of Eq. (19) is equal to $\dot{s}$ when $z = -\dot{s}$ and $\bar{j} = \dot{s}$, and hence the bound Eq. (19) is tight when $J = S$ and $z = \dot{s}$. The Gallavotti-Cohen fluctuation relation Eq. (60) also applies for currents $J$ that are proportional to the entropy production [46,47], and hence the bound Eq. (3) is also tight for those currents. Importantly, the fluctuation relation does not apply generically for currents in multicyclic networks that are not proportional to $S$ [46–49], and hence the inequality (3) is not tight for generic currents.

# 7 Recovering the Van't Hoff-Arrhenius law in the near equilibrium limit

We show that near equilibrium Eq. (3) implies that $1/\langle T_J \rangle$ is smaller or equal than the Van't Hoff-Arrhenius law Eq. (5). To this aim, we consider a nonequilibrium version of Kramers' model [1,3]. Details of the calculations can be found in the Appendices B and C.

We consider a reaction coordinate $X \in \mathbb{R}$ that is described by the overdamped Langevin equation

$$dX(t) = \frac{f - \partial_x u(X(t))}{\gamma} dt + \sqrt{2\mathsf{T}_{\text{env}}/\gamma} \, dW(t), \tag{61}$$

where $u(x)$ is a periodic potential with period $\delta$, i.e., $u(x + \delta) = u(x) = u(x - \delta)$, $f$ is a nonconservative force, $\gamma$ is a friction coefficient, $W(t)$ is a standard Wiener process that models the thermal noise, and $\mathsf{T}_{\text{env}}$ is the temperature of the environment. We assume that at time $t = 0$, $X(0) = 0$ and $W(0) = 0$. Note that this example goes beyond the pardigm of a Markov jump process, but the theory will still apply.

The variable $X$ models, e.g., a reaction coordinate that tracks the progress of a chemical reaction. In this scenario, $E_{\text{b}} = \max_x u(x) - \min_x u(x)$ is the Gibbs free energy barrier that separates two chemical states and the ratio $[X/\delta]$ is the number of cycles of the reaction that have been completed; $[a]$ denotes the largest integer smaller than $a$.

Figure 1 presents two trajectories generated by Eq. (61) for the special case where $u(x)$ is the triangular potential

$$u(x) = \begin{cases} u_0 \frac{x}{x^*} & \text{if} \quad x \in [0, x^*), \\ u_0 \frac{\delta - x}{\delta - x^*} & \text{if} \quad x \in [x^*, \delta). \end{cases} \tag{62}$$

From Fig. 1 we observe that the dynamics consists of a sequence of jumps between metastable states that are centred at the positions $nx^*$ with $n \in \mathbb{Z}$. In the equilibrium case with $f = 0$ the jumps are activated by thermal fluctuations and the Van't Hoff-Arrhenius law Eq. (5) applies. On the other hand, when $f > 0$, then jumps in one direction over the energy barrier $E_b$ are facilitated by the external driving $f$, while in the reverse direction jumps are less likely. In this case, although the Van't Hoff-Arrhenius law Eq. (5) does not apply, the Eqs. (3) and (6) apply and can thus be considered nonequilibrium versions of the Van't Hoff-Arrhenius law.

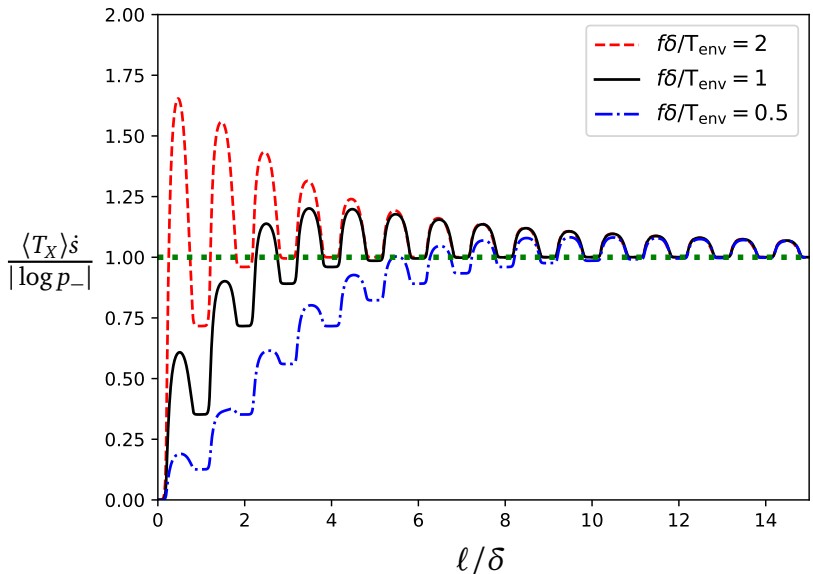

Figure 3: *Asymptotic lower bound on the mean first-passage time.* The ratio $\langle T_X \rangle \dot{s}/|\log p_-|$ is plotted as a function of $\ell/\delta$, where $T_X$ is the first-passsage time Eq. (2) of the nonequilibrium Kramer process $X$ described by Eq. (61) with triangular potential $u$ given by Eq. (62). Curves shown are for the parameters $\delta = 5$, $x^* = 1$, $u_0 = 10$, $\mathsf{T}_{env} = 1$, and $\gamma = 1$, and the values of $f$ are given in the figure legend.

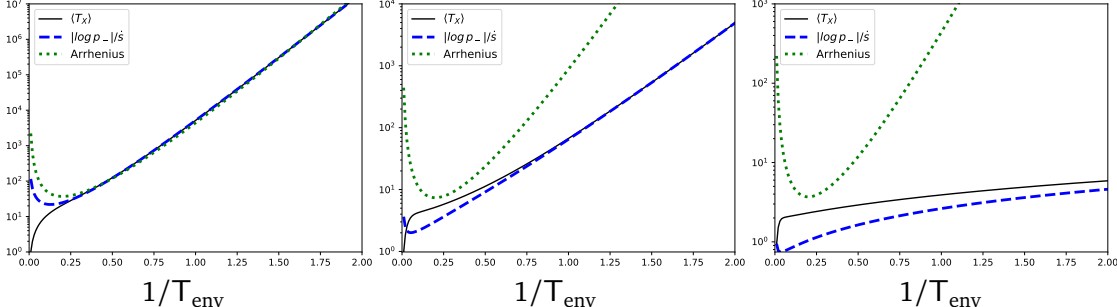

Figure 4: *Extension of the Van't Hoff-Arrhenius law to nonequilibrium stationary states.* The mean-first passage time $\langle T_X \rangle$ (solid black line) of the reaction coordinate $X$, described by Eq. (61) with triangular potential $u$ given by Eq. (62), is plotted as a function of the inverse temperature $1/\mathsf{T}_{env}$, and $\langle T_X \rangle$ is also compared with its asymptotic value $|\log p_-|/\dot{s}$ for large thresholds $\ell$ (blue dashed line) and with the Van't Hoff-Arrhenius law Eq. (72) (green dotted line). The model parameters are $\delta = 5$, $x^* = 1$, $u_0 = 10$, $\mathsf{T}_{env} = 1$ and $\gamma = 2$ and the values of $f$ are $f = 1$, $f = 5$ and $f = 10$ from left to right, respectively. The threshold for the first-passage time $T_X$, which is defined in Eq. (65), is $\ell = 10$.

For values $f\delta/E_{\rm b} > 0$ the chemical reaction settles into a nonequilibrium stationary state with an entropy production rate (see Appendix B.2)

$$\dot{s} = \frac{f\delta}{\mathsf{T}_{\rm env}} j_{\rm ss}, \tag{63}$$

where $j_{\rm ss}$ is the stationary current (see Appendix B.1)

$$j_{\rm ss} = \frac{\mathsf{T}_{\rm env}}{\gamma} \frac{1 - e^{\frac{-f\delta}{\mathsf{T}_{\rm env}}}}{\int_0^\delta \mathrm{d}y\, w(y) \left( \int_y^{y+\delta} \mathrm{d}x' \frac{1}{w(x')} \right)}, \tag{64}$$

and where $w(x) = \exp(-(u(x) - fx)/\mathsf{T}_{\rm env})$.

Consider the first time

$$T_X = \inf\{t > 0 : X(t) \notin (-\ell, \ell)\}, \tag{65}$$

when the reaction has completed a net number $[\ell/\delta]$ of cycles in either the forward or backward direction. Since, (see Appendix B.2)

$$S(t) = \frac{f X(t)}{\mathsf{T}_{\rm env}} + o(t), \tag{66}$$

the equality (6) applies to $T_X$. In Appendices B.3 and B.4, we derive explicit analytical expressions for the splitting probability $p_-$ and the mean first-passage time $\langle T_X \rangle$, respectively, which we omit here as the expressions are involved. However, as shown in Appendix B.5, in the limit of large $\ell$ we obtain the formula

$$\frac{|\log p_-|}{\langle T_X \rangle} = \dot{s} + O\left(\frac{1}{\ell}\right), \tag{67}$$

in correspondence with Eq. (6), where $O$ denotes the big-O notation. The big-O notation $O(f(\ell))$ denotes an arbitrary function $g(\ell)$ for which it holds that there exists a constant $c$ such that $g(\ell) < cf(\ell)$ for $\ell$ large enough. Hence, in this case, the correction term in Eq. (6) is of order $1/\ell$.

In Fig. 3 we plot $|\log p_-|\dot{s}/\langle T_X \rangle$ as a function of $\ell/\delta$. The figure demonstrates the convergence of $|\log p_-|\dot{s}/\langle T_X \rangle$ to its universal limit for different values of the nonequilibrium driving $f\delta/\mathsf{T}_{\rm env}$. Observe the oscillations of $|\log p_-|\dot{s}/\langle T_X \rangle$. These oscillations appear because for the selected parameters it holds that $E_{\rm b} \gg \mathsf{T}_{\rm env}$, and therefore the process consists of discrete-like hops over the energy barrier $E_{\rm b}$ that represent the subsequent completion cycles of the chemical reaction.

In the limits $\mathsf{T}_{\rm env} \to 0$ and $f\delta/\mathsf{T}_{\rm env} \to 0$, the Eq. (6) leads to a Van't Hoff-Arrhenius law for $1/\langle T_X \rangle$. Indeed, as shown in Appendix B.6, taking the limits $\mathsf{T}_{\rm env} \to 0$ and $f\delta/\mathsf{T}_{\rm env} \to 0$ in the expression of the stationary current Eq. (64), we obtain

$$j_{\rm ss} = \kappa \frac{f\delta}{\gamma} e^{\frac{-E_{\rm b}}{\mathsf{T}_{\rm env}}}, \tag{68}$$

where the prefactor

$$\kappa = \frac{\sqrt{-u''_{\min} u''_{\max}}}{2\pi \mathsf{T}_{\rm env}} \tag{69}$$

if the second derivatives $u''_{\min}$ and $u''_{\max}$ evaluated at the minimum and maximum of $u(x)$, respectively, exist. In the special case of the triangular potential, given by Eq. (62), the second derivatives $u''_{\min}$ and $u''_{\max}$ do not exist, and therefore

$$\frac{1}{\kappa} = \left( \frac{1}{u^+_{\max}} - \frac{1}{u^-_{\max}} \right) \left( \frac{1}{u^+_{\min}} - \frac{1}{u^-_{\min}} \right) \mathsf{T}^2_{\rm env}, \tag{70}$$

where $u_{\max}^+$ and $u_{\max}^-$ denote the left and right derivatives evaluated at the maximum of $u(x)$. In addition, as shown in Appendix B.6, in the limit of $T_{env} \to 0$ and $f\delta/T_{env} \to 0$ the logarithm of the splitting probability is inversely proportional to the temperature, viz.,

$$\log p_- = -\frac{f\ell}{T_{env}} + O_\ell(1). \tag{71}$$

Combining Eqs. (6), (63), (68), and (71) we obtain the Van't Hoff-Arrhenius law

$$\langle T_X \rangle = \frac{\ell}{\delta} \frac{\gamma}{f\delta} \frac{1}{\kappa} e^{\frac{E_b}{T_{env}}}. \tag{72}$$

In Fig. 4 we compare $\langle T_X \rangle$ with its asymptotic value $|\log p_-|/\dot{s}$, given by Eq. (6), and with the Van't Hoff-Arrhenius law, given by Eq. (72), for three values of the driving force $f$. We make a few interesting observations: (i) the Van't Hoff-Arrhenius law approximates well $\langle T_X \rangle$ up to moderately large values of $f\delta/T_{env} < 5$; (ii) for $f\delta/T_{env} > 25$, $\langle T_X \rangle$ is significantly smaller than what is predicted by the Van't Hoff-Arrhenius law, implying that the nonequilibrium driving speeds up the process. Nevertheless, $\langle T_X \rangle$ is larger than $|\log p_-|/\dot{s}$, which is a consequence of the trade-off between speed, uncertainty, and dissipation as expressed by Eq. (3); (iii) the asymptotic expression $|\log p_-|/\dot{s}$ given by Eq. (6) approximates $\langle T_X \rangle$ already well for relatively small values of the threshold, viz., $\ell/\delta = 2$.

Taken together, we conclude that the Eqs. (3) and (6) recover the Van't Hoff-Arrhenius law near equilibrium because $\dot{s} \sim \exp(-E_b/T_{env})$ in the limit of small temperatures $T_{env} \approx 0$ and small driving force $f\delta/T_{env} \approx 0$. On the other hand, one can can significantly increase the reaction rate $1/\langle T_X \rangle$ by driving a system out of equilibrium, even though the reaction rates are still bounded from above by the inequality Eq. (3) that expresses a tradeoff between speed, uncertainty, and dissipation.

## 8 Illustration of the tightness of the first-passage time bounds with a biased random walker

As stated before, the bound Eq. (3) is tight for $J = S$, whereas the thermodynamic uncertainty relation Eq. (59) is loose when $J = S$. In this section we compute the moments $\langle T_J^n \rangle$ for an example of a nonequilibrium process to better understand the origin of the tightness of the bound Eq. (3).

We consider a hopping process $X \in \mathbb{Z}$ described by

$$dX(t) = dN_+(t) - dN_-(t), \tag{73}$$

where $N_+$ and $N_-$ are two counting process with rates $k_+$ and $k_-$, respectively. The bias of the process is defined by the ratio

$$b := \frac{k_-}{k_+} = \exp\left(-\frac{a}{T_{env}}\right), \tag{74}$$

where $a$ is the thermodynamic affinity and $T_{env}$ the temperature of the environment. We assume, without loss of generality, that $k_- < k_+$ so that $b < 1$.

The coordinate $X$ may represent the number of times a chemical reaction has been completed or the position of a molecular motor on a biofilament. In the former, $a = \Delta\mu$ is the difference between the sum of the chemical potentials of the reagents and the products of

the chemical reaction, and in the latter $a = f\delta$ is the work performed by the motor on its environment. Hence, the stochastic entropy production $S$ obeys

$$dS(t) = \frac{a}{\mathsf{T}_{\text{env}}} dX(t) \tag{75}$$

and

$$\dot{s} = \left\langle \frac{dS}{dt} \right\rangle = \frac{a}{\mathsf{T}_{\text{env}}}(k_+ - k_-) \tag{76}$$

is the entropy production rate.

We consider the first passage time

$$T_X = \inf\{t > 0 : X(t) - X(0) \notin (-\ell_-, \ell_+)\}, \tag{77}$$

which is also the first-passage time $T_S$ for the stochastic entropy production with thresholds $s_- = a\ell_-/\mathsf{T}_{\text{env}}$ and $s_+ = a\ell_+/\mathsf{T}_{\text{env}}$.

The splitting probabilities $p_-$ and $p_+$ are given by (see Appendix D.3)

$$p_+ = \frac{1 - b^{[\ell_-]}}{1 - b^{[\ell_-] + [\ell_+]}} \quad \text{and} \quad p_- = b^{[\ell_-]} \frac{1 - b^{[\ell_+]}}{1 - b^{[\ell_-] + [\ell_+]}}, \tag{78}$$

where $[\ell_-]$ and $[\ell_+]$ denote the largest integers that are smaller than $\ell_-$ and $\ell_+$, respectively. The generating function

$$g(y) = \langle e^{-yT_X(k_- + k_+)} \rangle \tag{79}$$

is for all $y > 0$ given by (see Appendix D.4)

$$g(y) = \left(\frac{2}{\zeta_+(y)}\right)^{[\ell_+]} \frac{1 - \left(\frac{\zeta_-(y)}{\zeta_+(y)}\right)^{[\ell_-]}}{1 - \left(\frac{\zeta_-(y)}{\zeta_+(y)}\right)^{[\ell_-] + [\ell_+]}} + \left(\frac{\zeta_-(y)}{2}\right)^{[\ell_-]} \frac{1 - \left(\frac{\zeta_-(y)}{\zeta_+(y)}\right)^{[\ell_+]}}{1 - \left(\frac{\zeta_-(y)}{\zeta_+(y)}\right)^{[\ell_-] + [\ell_+]}}, \tag{80}$$

where

$$\zeta_\pm(y) = \beta(y) \pm \sqrt{-4b + \beta^2(y)} \tag{81}$$

and

$$\beta(y) = (1 + y)(1 + b). \tag{82}$$

The moments of $T_X$ follow from

$$\langle T_X^n \rangle = \left(\frac{-1}{k_- + k_+}\right)^n \frac{d^n}{(dy)^n} g(y) \bigg|_{y=0}, \tag{83}$$

where $n \in \mathbb{N}$.

Figure 5 compares the first-passage time bounds Eqs. (3) with the thermodynamic uncertainty relation Eq. (59). The plotted curves are obtained from the explicit analytical expressions for $\dot{s}$ and $p_-$, given by Eqs. (76) and (78), respectively, and from explicit analytical expressions for $\langle T^n \rangle$ that we have obtained from the Eqs. (79-83) and can be found in the Appendix D.6. The figure shows that for large values of the first-passage thresholds the bounds Eqs. (3) are tight, as predicted by Eq. (6), while the thermodynamic uncertainty relation is loose.

In Fig. 5 we also observe that the first moment $\langle T \rangle$ converges fast to its asymptotic value, while higher order moments $\langle T^2 \rangle$ and $\langle T^3 \rangle$ converge slowly to their asymptotic values. Using Eqs. (76), (78), and (79-83), we obtain the asymptotics (see Appendices D.7 and D.8)

$$\frac{[\ell_+]}{[\ell_-]} \frac{|\log p_-|}{\langle T_X \rangle} = \dot{s} + O\left(b^{[\ell_-]}\right), \tag{84}$$



and for $n > 1$

$$\frac{[\ell_+]}{[\ell_-]}\frac{|\log p_-|}{\left(\langle T_X^n\rangle\right)^{1/n}} = \dot{s} + O\left(\frac{1}{[\ell_+]}\right). \tag{85}$$

Hence, the first moment converges exponentially fast to the entropy production rate $\dot{s}$, while the higher order moments converge as $1/[\ell_+]$ to their asymptotic value. Consequently, in this example the first moment is more effective for the inference of the entropy production rate $\dot{s}$. However, from Eq. (67) we can conclude that the exponential fast convergence for the first moment is a model specific property.

The asymptotic expression for the thermodynamic uncertainty relation depends on the subleading $O(1/[\ell_+])$ term in Eq. (85), and is given by

$$\frac{2\langle T_X\rangle}{\langle T_X^2\rangle - \langle T_X\rangle^2} = \frac{2(k_+ - k_-)}{\tanh\left(\frac{a}{2\mathsf{T}_{\text{env}}}\right)} + O\left(b^{[\ell_-]}\right). \tag{86}$$

Since $\tanh(x) \le x$, the thermodynamic uncertainty relation Eq. (59) holds. However, contrary to Eqs. (84) and (85), the thermodynamic uncertainty relation is not tight in the limit of large thresholds and the ratio Eq. (86) depends on the affinity $a/\mathsf{T}_{\text{env}}$ of the process.

Taken together, we can conclude that the equality Eq. (6), and thus the tightness of the bound Eq. (3) for $J = S$, follows from the universality of the leading order term in the Eqs. (84) and (85) for $\langle T_X^n\rangle$. On the other hand, the looseness of the thermodynamic uncertainty relation Eq. (59) for $S = J$ is a consequence of the nonuniversality of the subleading order terms of $\langle T_X\rangle$ and $\langle T_X^2\rangle$ in the Eqs. (84) and (85) and therefore the right-hand side of Eq. (86) depends on the affinity $a$ of the process.

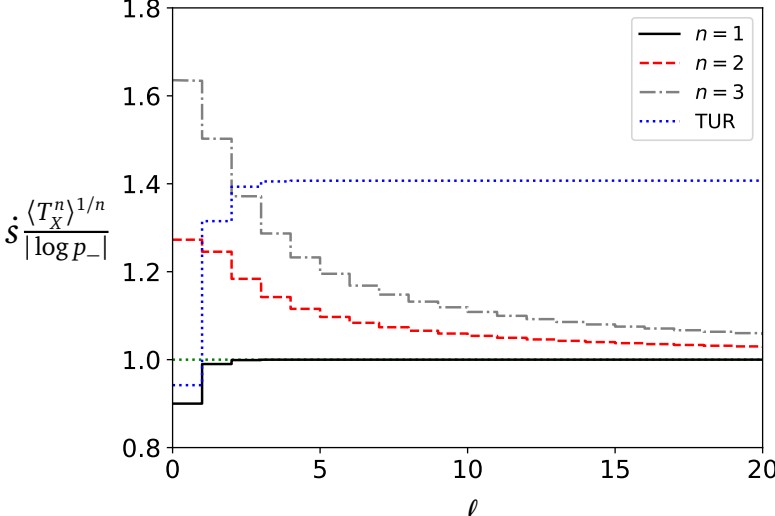

Figure 5: *Comparing the tightness of the first-passage time bounds Eq. (3) with the thermodynamic uncertainty relation Eq. (59). The ratio $\dot{s}\langle T_X^n\rangle^{1/n}/|\log p_-|$ for $n = 1, 2, 3$ and the thermodynamic uncertainty (TUR) ratio $\dot{s}(\langle T_X^2\rangle - \langle T_X\rangle^2)/(2\langle T_X\rangle)$ are plotted as a function of $\ell = \ell_- = \ell_+$ for a biased random walk process $X$ described by Eq. (73) with $k_+ = 1$ and $b = 0.1$. Note that the inequalities Eq. (3) are tight for $\ell \to \infty$, while the uncertainty relation Eq. (59) is loose.*

## 9  Discussion

Driving a system out of equilibrium can speed up the rate of a chemical reaction. However, there exists a fundamental thermodynamic tradeoff between speed, the fluctuations in the process, and the rate of dissipation. The main contribution of this paper is the derivation of a universal inequality, Eq. (3), that expresses in nonequilibrium stationary states a thermodynamic tradeoff between speed, uncertainty, and dissipation, which are quantified in terms of the mean first passage time $\langle T_J \rangle$, the splitting probability $p_-$, and the dissipation rate $\dot{s}$, respectively. The main advantage of the inequality (3) with respect to previously published trade-off relations, such as the thermodynamic uncertainty relations [32,33,43,50–56], is that Eq. (3) is an equality when $J(t) = cS(t)$ with $c$ a time and trajectory independent constant, see Eq. (6), and hence the bound is optimal in this case.

From a physical and mathematical point of view, the Eqs. (3) and (6) are interesting as they are related to thermodynamic uncertainty relations, the Van't Hoff-Arrhenius law, martingale theory, and the theory of sequential hypothesis testing. Indeed, both Eq. (3) and the thermodynamic uncertainty relations are a consequence of the large deviation function bound Eq. (19). On the other hand, the equality Eq. (6) follows from martingale theory [20,21], in particular the integral fluctuation relation at stopping times [23]. We have also recovered the Van't Hoff-Arrhenius law Eq. (5) in the near equilibrium limit $\dot{s} \to 0$. In addition, we have also derived Eqs. (3) and (6) from the theory of sequential hypothesis testing [37,40], more specifically, the asymptotic optimality of sequential probability ratio tests. It is fascinating that all these different research areas are related to each other and certainly more fundamental insights about stochastic thermodynamics can be gained by exploring the links between these areas.

The present paper derives the main result Eq. (3) in the setup of currents $J$ in stationary Markov jump processes $X$; in addition, to identify $\dot{s}$, as defined in Eq. (11)-(13), with the mean rate of dissipation we require local detailed balance. Nevertheless, we expect that (3) can be generalised. In Sec. 4, we have derived the bound (3) using large deviation theory, in particular, we have used the bound (19) on the large deviation function of the current. Since the bound (19) has been derived for stationary Markov jump processes, see Ref. [33], also (3) applies to this setup. Consequently, (3) extends to processes $X$ for which a bound on the large deviation function of the form (19) holds. Notable examples worthwhile exploring are overdamped Langevin processes [57] and asymptotically stationary processes with time-dependent driving [58]. Another possible avenue of approach for generalising (3) is based on the theory of sequential hypothesis testing, as presented in Sec. 5. In this approach, we have derived the partial result (47) in a very general setup and with full mathematical rigour. However, to get (3) we relied on the additional result (48), which has not been derived with the same level of mathematical rigour as (47). It will be interesting to establish the conditions under which (48) holds with full mathematical rigour, as this will pave the way for extensions beyond the setup of stationary Markov jump processes.

The equality (6) has been derived in more general setup than the bound (3). In Sec. 5, we have presented a rigorous derivation of (3) based on the martingale property of $e^{-S}$ and the $r$-quick convergence of $S(t)/t$ to a deterministic limit. The martingale property of $e^{-S}$ holds as long as it can be written in the form [20,21,23,59–61]

$$e^{-S(t)} = \frac{\tilde{p}(\vec{X}_0^t)}{p(\vec{X}_0^t)},\tag{87}$$

with $\tilde{p}$ a probability distribution characterising the statistics of trajectories in the time-reversed process, whereas the $r$-quick convergence is a mild condition on the fluctuations of $S(t)/t$ in the limit of large $t$. In Langevin processes, including nonstationary processes, $e^{-S}$ can be

written in the form (87), see e.g. Refs. [19, 60], and hence the equality (6) should also apply to continuous stochastic processes.

We end the paper with a brief discussion of potential applications for the Eqs. (3) and (6). The inequality Eq. (3) could be used to infer dissipation rates from the measurements of first-passage times of stochastic currents. It is difficult to measure the entropy production rate directly as it is related to the heat exchanged with the environment [62]. However, since the mean first-passage time $\langle T_J \rangle$ and the splitting probability $p_-$ are directly measurable quantities, Eq. (3) can be used to bound the entropy production rate from below. When compared with other methods that infer entropy production rates from the measurements of stochastic currents, see e.g. [63–67], the present inequalities may turn out to perform better as they are optimal when $J = S$, although this requires further study as the inequality Eq. (3) has also some drawbacks. In particular, the probability $p_-$ decreases exponentially with $\ell_-$, which raises the question how $p_-$ can be estimated at large values of $\ell_-$. A second interesting application is in the study of first-passage problems of nonequilibrium processes, such as, those of self-propelled particles [4–8]. The inequality (3) and the equality (6) are generic results with a clear physical meaning, and hence, when used to bound the statistics of first-passage problems in nonequilibrium processes, can provide further physical understanding of mathematical results. A third interesting application is in the use of the bound Eq. (3) to determine how far molecular systems operate from what is physically nonpermissible. Notable examples are molecular motors that are responsible for copying genetic information in biological cells, such as, ribosomes or polymerases. These motors are known to attain a reliability that is larger than what is possible in equilibrium through kinetic proof reading [68–70], but it is not known how close to the physically nonpermissible limits these motors operate. Another example are transistors that are small enough so that they are prone to noise [71]. Bounds of the form Eq. (3) could be used to understand thermodynamic limitations on computing that are based on the tradeoff between dissipation, speed, and uncertainty in nonequilibrium processes.

## Acknowledgements

The author thanks Andre Barato, Patrick Pietzonka, and Benjamin Walter for insightful discussions.

## A  Martingales

In this appendix, we state the definition of a martingale and one of its key properties that we use repeatedly in this paper, namely, Doob's optional stopping theorem.

### A.1  Definition of a martingale

Let $\Omega$ be the set of all realisations of a physical process $X$, which is endowed with a $\sigma$-algebra $\mathscr{F}$. Let $\mathbb{P}$ be a probability measure that determines the probabilities $\mathbb{P}[\Phi]$ of events $\Phi \in \mathscr{F}$. We denote averages with respect to $\mathbb{P}$ by $\langle \cdot \rangle$. Let $\{\mathscr{F}(t)\}_{t \geq 0}$ be the filtration generated by $X$, i.e., a sequence of sub-$\sigma$-algebras $\mathscr{F}(t)$ that is generated by the trajectories $X_0^t$ of the process $X$.

A martingale $M(t)$ with respect to a filtration $\{\mathscr{F}(t)\}_{t \geq 0}$ is a stochastic process for which (i) the process $M(t)$ is $\mathscr{F}(t)$-measurable (ii) $\langle |M(t)| \rangle < \infty$ (iii) $\langle M(t)|\mathscr{F}(s) \rangle = M(s)$ [72, 73]. The latter condition implies that the martingale $M$ is a driftless process.

## A.2 Doob's optional stopping theorem

A stopping time $T$ is a random time $T : \Omega \to \mathbb{R}^+ \cup \{\infty\}$ such that $\{T \leq t\} \in \mathscr{F}(t)$ for all values of $t \in \mathbb{R}^+$. This means that $T$ stops the process $X$ based on a stopping rule that does not anticipate the future or use side information.

One of the key properties of martingales that we use in this paper is described by Doob's optional stopping theorem [73].

**Theorem 2** (Doob's optional stopping theorem). *Let $(\Omega, \mathscr{F}, \mathbb{P})$ be a probability space with sample space $\Omega$, $\sigma$-algebra $\mathscr{F}$, and probability measure $\mathbb{P}$. Let $X(t)$ with $t \geq 0$ be a $\mathscr{F}$-measurable stochastic process and let $\{\mathscr{F}(t)\}_{t \geq 0}$ be the filtration generated by $X$. Let $M$ be a martingale process with respect to the filtration $\{\mathscr{F}(t)\}_{t \geq 0}$ and let $T$ be a stopping time relative to the filtration $\{\mathscr{F}(t)\}_{t \geq 0}$. It holds then that*

$$\langle M(T \wedge t) \rangle = \langle M(0) \rangle, \tag{88}$$

*where $T \wedge t = \min\{T, t\}$.*

# B  Mean first-passage time for an overdamped Brownian particle in a generic periodic potential and in a uniform force field

In this appendix, we analyse the first-passage problem for a Brownian motion in a generic periodic potential $u$ and a uniform force field $f$, as described by Eq. (61). In particular, we derive analytical expressions for the mean first-passage time $\langle T_X \rangle$, the splitting probability $p_-$, and the mean entropy production rate $\dot{s}$, where $T_X$ is defined as in Eq. (65). In the limit of large thresholds $\ell_- = \ell_+ = \ell \gg 1$, we show that the main result Eq. (6) holds. In addition, in the near-equilibrium limit and at low temperatures, we show that Eq. (6) is a Van't Hoff-Arrhenius law.

## B.1  Stationary distribution and current

We derive Eq. (64) in the main text for the stationary current $j_{\text{ss}}$.

The stationary distribution of $X \in \mathbb{R}$ does not exist. However, we can define the process on a ring with periodic boundary conditions such that $X(t) = X(t) + \delta$. The stationary state $p_{\text{ss}}$ of the equivalent process defined on a ring exists, and we can use this process on a ring to determine the stationary current $j_{\text{ss}}$.

The stationary distribution $p_{\text{ss}}$ solves the equation [19, 74]

$$\partial_x j_{\text{ss}}(x) = 0 \tag{89}$$

with periodic boundary conditions $p_{\text{ss}}(x) = p_{\text{ss}}(x + \delta)$, where

$$j_{\text{ss}}(x) = \mu(f - \partial_x u(x)) p_{\text{ss}}(x) - \frac{\mathsf{T}_{\text{env}}}{\gamma} \partial_x p_{\text{ss}}(x). \tag{90}$$

The solution to Eq. (89) is given by [23, 75]

$$p_{\text{ss}}(x) = \frac{w(x)\left(\int_x^{x+\delta} \mathrm{d}x' \frac{1}{w(x')}\right)}{\int_0^\delta \mathrm{d}y\, w(y)\left(\int_y^{y+\delta} \mathrm{d}x' \frac{1}{w(x')}\right)}, \tag{91}$$

with $x \in [0, \delta]$, and where

$$w(x) = e^{-\frac{u(x) - fx}{\mathsf{T}_{\text{env}}}}. \tag{92}$$

The expression Eq. (64) for the stationary current $j_{\text{ss}}$ follows readily from the Eqs. (90) and (91).

## B.2 Entropy production

We derive Eqs. (63) and (66) in the main text for the entropy production rate $\dot{s}$ and the stochastic entropy production $S$, respectively. We will again use the equivalent process defined on a ring with periodic boundary conditions.

The stochastic entropy production $S$ of $X$, as defined in Eq. (11), is determined by the stochastic differential equation [22, 76]

$$dS = v_S(X)\,dt + \sqrt{2v_S(X)}\,dW(t)\,, \tag{93}$$

where

$$v_S(x) = \frac{\gamma}{\mathsf{T}_{\text{env}}}\frac{j_{\text{ss}}^2}{p_{\text{ss}}^2(x)} = \frac{\mathsf{T}_{\text{env}}}{\gamma}\frac{\left(1 - e^{\frac{-f\delta}{\mathsf{T}_{\text{env}}}}\right)^2}{w^2(x)\left(\int_x^{x+\delta}dx'\frac{1}{w(x')}\right)^2}\,. \tag{94}$$

Alternatively, we can write

$$S(t) = \frac{f X(t) - u(X(t)) + u(X(0))}{\mathsf{T}_{\text{env}}} + \log\frac{p_{\text{ss}}(X(0))}{p_{\text{ss}}(X(t))}\,. \tag{95}$$

The latter formula implies for large $t \gg 1$ that

$$S(t) = \frac{f X(t)}{\mathsf{T}_{\text{env}}} + o(t)\,, \tag{96}$$

which is Eq. (66) in the main text.

The average stationary entropy production rate is given by

$$\dot{s} = \frac{\langle S(t)\rangle}{t} = \langle v_S\rangle = \frac{\gamma j_{\text{ss}}^2}{\mathsf{T}_{\text{env}}}\int_0^\delta \frac{dx}{p_{\text{ss}}(x)}\,. \tag{97}$$

Since the stationary distribution $p_{\text{ss}}$ is given by Eq. (91) and $u(x)$ is a periodic function, we can express this also as

$$\dot{s} = j_{\text{ss}}\left(1 - e^{\frac{-f\delta}{\mathsf{T}_{\text{env}}}}\right)\int_0^\delta dx\,\frac{1}{w(x)\left(\int_0^\delta dx'\frac{1}{w(x')} - (1 - e^{-\frac{f\delta}{\mathsf{T}_{\text{env}}}})\int_0^x dx'\frac{1}{w(x')}\right)}\,. \tag{98}$$

Introducing the function

$$\int_0^x dx'\frac{1}{w(x')} = W(x)\,, \tag{99}$$

we find that

$$\dot{s} = j_{\text{ss}}\left(1 - e^{\frac{-f\delta}{\mathsf{T}_{\text{env}}}}\right)\int_0^{W(\delta)} du\,\frac{1}{W(\delta) - (1 - e^{-\frac{f\delta}{\mathsf{T}_{\text{env}}}})u}\,. \tag{100}$$

Integrating yields the expression for $\dot{s}$ given by Eq. (63) in the main text.

## B.3 Splitting probabilities

We use the martingale property of $e^{-S(t)}$, see Refs. [21, 23] or Appendix A, to determine the splitting probabilities $p_-$ and $p_+$. Doob's optional stopping theorem for martingales implies the following integral fluctuation relation at stopping times

$$\langle e^{-S(T_X)}|X(0) = 0\rangle = e^{-S(0)} = 1\,, \tag{101}$$

and since $S(t)$ is continuous as a function of $t$ this implies that, see Refs. [21, 23],

$$p_- = e^{-s_-} \frac{1 - e^{-s_+}}{1 - e^{-s_- - s_+}}, \quad \text{and} \quad p_+ = \frac{1 - e^{-s_-}}{1 - e^{-s_- - s_+}}, \tag{102}$$

where

$$s_- = -\frac{-f\ell - u(-\ell) + u(0)}{\mathsf{T}_{\text{env}}} - \log \frac{p_{\text{ss}}(0)}{p_{\text{ss}}(-\ell)}, \quad \text{and} \quad s_+ = \frac{f\ell - u(\ell) + u(0)}{\mathsf{T}_{\text{env}}} + \log \frac{p_{\text{ss}}(0)}{p_{\text{ss}}(\ell)}. \tag{103}$$

Notice that we have used a slight abuse of notation in the sense that $u(x)$ and $p_{\text{ss}}(x)$ are here defined on $x \in \mathbb{R}$ using $u(x) = u(x \pm \delta)$ and $p_{\text{ss}}(x) = p_{\text{ss}}(x \pm \delta)$.

## B.4 Mean first-passage time

Consider the backward Fokker-Planck equation

$$\mu(f - \partial_x u(x)) \partial_x t(x) + \frac{\mathsf{T}_{\text{env}}}{\gamma} \partial_x^2 t(x) = -1, \tag{104}$$

with boundary conditions $t(-\ell) = t(\ell) = 0$. It then holds that, see Ref. [77],

$$\langle T_X | X(0) = x \rangle = t(0). \tag{105}$$

The solution of $t(x)$ to Eq. (104) with boundary conditions $t(-\ell) = t(\ell) = 0$ is given by

$$t(x) = \frac{\gamma}{\mathsf{T}_{\text{env}}} \left( \int_{-\ell}^{\ell} dy \frac{1}{w(y)} \int_0^y dx' w(x') \right) \left( \frac{\int_{-\ell}^x dy \frac{1}{w(y)}}{\int_{-\ell}^{\ell} dy \frac{1}{w(y)}} - \frac{\int_{-\ell}^x dy \frac{1}{w(y)} \int_0^y dx' w(x')}{\int_{-\ell}^{\ell} dy \frac{1}{w(y)} \int_0^y dx' w(x')} \right), \tag{106}$$

and therefore

$$\langle T_X \rangle = \frac{\gamma}{\mathsf{T}_{\text{env}}} \left( \int_{-\ell}^{\ell} dy \frac{1}{w(y)} \int_0^y dx' w(x') \right) \left( \frac{\int_{-\ell}^0 dy \frac{1}{w(y)}}{\int_{-\ell}^{\ell} dy \frac{1}{w(y)}} - \frac{\int_{-\ell}^0 dy \frac{1}{w(y)} \int_0^y dx' w(x')}{\int_{-\ell}^{\ell} dy \frac{1}{w(y)} \int_0^y dx' w(x')} \right). \tag{107}$$

In order to better understand the structure of the expression Eq. (107) for the mean-first passage time, it is useful to express the integrals in Eq. (107) that run over the intervals $[-\ell, \ell]$ and $[-\ell, 0]$ in terms of integrals that run over the interval $[0, \delta]$. Let $n = [\ell/\delta]$ be the largest integer smaller than $\ell/\delta$, then we can write

$$\ell = n\delta + z, \tag{108}$$

with $z \in [0, \delta]$. Using this decomposition for $\ell$, we obtain that

$$\int_{-n\delta-z}^0 dy \frac{1}{w(y)} = e^{n \frac{f\delta}{\mathsf{T}_{\text{env}}}} \left\{ \left( \frac{1 - e^{-n \frac{f\delta}{\mathsf{T}_{\text{env}}}}}{1 - e^{-\frac{f\delta}{\mathsf{T}_{\text{env}}}}} \right) \int_0^\delta \frac{dx}{w(x)} + e^{\frac{f\delta}{\mathsf{T}_{\text{env}}}} \int_{\delta-z}^\delta \frac{dx}{w(x)} \right\} \tag{109}$$

and

$$\int_{-n\delta-z}^{n\delta+z} dy \frac{1}{w(y)}$$
$$= e^{n \frac{f\delta}{\mathsf{T}_{\text{env}}}} \left\{ \left( \frac{1 - e^{-2n \frac{f\delta}{\mathsf{T}_{\text{env}}}}}{1 - e^{-\frac{f\delta}{\mathsf{T}_{\text{env}}}}} \right) \int_0^\delta \frac{dx}{w(x)} + e^{\frac{f\delta}{\mathsf{T}_{\text{env}}}} \int_{\delta-z}^\delta \frac{dx}{w(x)} + e^{-2n \frac{f\delta}{\mathsf{T}_{\text{env}}}} \int_0^z \frac{dx}{w(x)} \right\}. \tag{110}$$

In addition,

$$\int_0^{n\delta+z} dy\, \frac{1}{w(y)} \int_0^y dx'\, w(x')$$

$$= n\left\{ \frac{e^{-\frac{f\delta}{T_{env}}}}{1-e^{-\frac{f\delta}{T_{env}}}} \int_0^\delta dx\, \frac{1}{w(x)} \int_0^\delta dx\, w(x) + \int_0^\delta dy\, \frac{1}{w(y)} \int_0^y w(x)dx \right\}$$

$$- \frac{e^{-\frac{f\delta}{T_{env}}}\left(1-e^{-n\frac{f\delta}{T_{env}}}\right)}{\left(1-e^{-\frac{f\delta}{T_{env}}}\right)^2} \int_0^\delta dx\, w(x) \int_0^\delta dx\, \frac{1}{w(x)}$$

$$+ e^{-\frac{f\delta}{T_{env}}}\frac{1-e^{-n\frac{f\delta}{T_{env}}}}{1-e^{-\frac{f\delta}{T_{env}}}} \int_0^z dy\, \frac{1}{w(y)} \int_0^\delta dx\, w(x) + \int_0^z dy\, \frac{1}{w(y)} \int_0^y dx\, w(x), \quad (111)$$

and

$$-\int_{-n\delta-z}^0 dy\, \frac{1}{w(y)} \int_0^y dx'\, w(x')$$

$$= \frac{1-e^{n\frac{f\delta}{T_{env}}}}{(1-e^{-\frac{f\delta}{T_{env}}})(1-e^{\frac{f\delta}{T_{env}}})} \left(\int_0^\delta dx\, w(x)\right)\left(\int_0^\delta dx\, \frac{1}{w(x)}\right)$$

$$+ n\left\{ \int_0^\delta dy\, \frac{1}{w(y)} \int_y^\delta dx\, w(x) - \frac{1}{1-e^{-\frac{f\delta}{T_{env}}}}\left(\int_0^\delta dx\, w(x)\right)\left(\int_0^\delta dx\, \frac{1}{w(x)}\right) \right\}$$

$$+ \frac{e^{n\frac{f\delta}{T_{env}}}-1}{1-e^{-\frac{f\delta}{T_{env}}}} \int_{\delta-z}^\delta dx\, \frac{1}{w(x)} \int_0^\delta dx\, w(x) + \int_{\delta-z}^\delta dy\, \frac{1}{w(y)} \int_y^\delta dx\, w(x). \quad (112)$$

Using the Eqs. (109), (110), (111), and (112) in Eq. (107), we obtain an expression for $\langle T_X \rangle$ that depends only on integrals over the interval $[0, \delta]$.

## B.5  Limit of large thresholds

We derive the Eq. (67) that holds in the limit of large $\ell$. The derivation goes in three steps. First, in Sec. B.5.1 we derive an asymptotic expression for $p_-$, second in Sec. B.5.2 we derive an asymptotic expression for $\langle T_X \rangle$, lastly in Sec. B.5.3 we combine these two results to determine the ratio $p_-/\langle T_X \rangle$.

### B.5.1  Splitting probabilities

In the limit of large thresholds, the linear term in $\ell$ dominates the Eqs. (103) and therefore

$$s_- = \frac{f\ell}{T_{env}} + O_\ell(1), \quad \text{and} \quad s_+ = \frac{f\ell}{T_{env}} + O_\ell(1). \tag{113}$$

Using Eq. (113) in the Eqs. (102) for $p_-$ and $p_+$, we obtain that

$$\log p_- = -\frac{f\ell}{T_{env}} + O_\ell(1), \quad \text{and} \quad \log p_+ = O_\ell(1). \tag{114}$$

### B.5.2  Mean first-passage time

We use that

$$n = \left[\frac{\ell}{\delta}\right] + O_\ell(1), \tag{115}$$

where as before $\left[\frac{\ell}{\delta}\right]$ denotes the largest integer that is smaller than $\frac{\ell}{\delta}$.

Taking the asymptotic limit of large $\ell$ in Eqs. (109) and (110), we obtain that

$$\frac{\int_{-\ell}^{0} dy \frac{1}{w(y)}}{\int_{-\ell}^{\ell} dy \frac{1}{w(y)}} = 1 - e^{-\left[\frac{\ell}{\delta}\right]\frac{f\delta}{T_{\text{env}}}} \frac{\int_{0}^{\delta} \frac{dx}{w(x)}}{\int_{0}^{\delta} \frac{dx}{w(x)} + (e^{\frac{f\delta}{T_{\text{env}}}} - 1)\int_{\delta-z}^{\delta} \frac{dx}{w(x)}} + O\left(e^{-2\left[\frac{\ell}{\delta}\right]\frac{f\delta}{T_{\text{env}}}}\right). \quad (116)$$

The asymptotic limit of Eq. (111) is

$$\int_{0}^{\ell} dy \frac{1}{w(y)} \int_{0}^{y} dx' w(x')$$

$$= \left[\frac{\ell}{\delta}\right]\left\{\frac{e^{-\frac{f\delta}{T_{\text{env}}}}}{1 - e^{-\frac{f\delta}{T_{\text{env}}}}} \int_{0}^{\delta} dx \frac{1}{w(x)} \int_{0}^{\delta} dx w(x) + \int_{0}^{\delta} dy \frac{1}{w(y)} \int_{0}^{y} w(x) dx\right\} + O_{\ell}(1), \quad (117)$$

and from Eqs. (111) and (112) it follows that

$$-\int_{-\ell}^{\ell} dy \frac{1}{w(y)} \int_{0}^{y} dx' w(x')$$

$$= e^{\left[\frac{\ell}{\delta}\right]\frac{f\delta}{T_{\text{env}}}} \left\{\frac{\int_{0}^{\delta} dx \, w(x) \int_{0}^{\delta} dx \frac{1}{w(x)}}{(1 - e^{-\frac{f\delta}{T_{\text{env}}}})(e^{\frac{f\delta}{T_{\text{env}}}} - 1)} + \frac{\int_{\delta-z}^{\delta} dx \frac{1}{w(x)} \int_{0}^{\delta} dx w(x)}{1 - e^{-\frac{f\delta}{T_{\text{env}}}}}\right\}$$

$$+ \left[\frac{\ell}{\delta}\right]\left\{\int_{0}^{\delta} dy \frac{1}{w(y)} \int_{y}^{\delta} dx \, w(x) - \frac{1}{\tanh\left(\frac{f\delta}{2T_{\text{env}}}\right)} \int_{0}^{\delta} dx \frac{1}{w(x)} \int_{0}^{\delta} dx \, w(x)\right.$$

$$\left. -\int_{0}^{\delta} dy \frac{1}{w(y)} \int_{0}^{y} dx \, w(x)\right\} + O_{\ell}(1). \quad (118)$$

The Eqs. (117) and (118) imply that the ratio

$$\frac{\int_{-\ell}^{0} dy \frac{1}{w(y)} \int_{0}^{y} dx' w(x')}{\int_{-\ell}^{\ell} dy \frac{1}{w(y)} \int_{0}^{y} dx' w(x')} = 1 + \left[\frac{\ell}{\delta}\right] e^{-\left[\frac{\ell}{\delta}\right]\frac{f\delta}{T_{\text{env}}}} \quad (119)$$

$$\times \left\{\frac{e^{-\frac{f\delta}{T_{\text{env}}}}\left(\int_{0}^{\delta} dx \, w(x)\right)\left(\int_{0}^{\delta} dx \frac{1}{w(x)}\right) + \left(1 - e^{-\frac{f\delta}{T_{\text{env}}}}\right)\int_{0}^{\delta} dy \frac{1}{w(y)} \int_{0}^{y} w(x) dx}{\frac{\int_{0}^{\delta} dx w(x) \int_{0}^{\delta} dx \frac{1}{w(x)}}{e^{\frac{f\delta}{T_{\text{env}}}} - 1} + \int_{\delta-z}^{\delta} dx \frac{1}{w(x)} \int_{0}^{\delta} dx \, w(x)}\right\} + O\left(e^{-\left[\frac{\ell}{\delta}\right]\frac{f\delta}{T_{\text{env}}}}\right).$$

Using Eqs. (116)-(119) in Eq. (107) yields for the mean first-passage time the asymptotic expression

$$\langle T_X \rangle = \frac{\gamma}{T_{\text{env}}} \left[\frac{\ell}{\delta}\right]\left[\frac{e^{-\frac{f\delta}{T_{\text{env}}}}}{1 - e^{-\frac{f\delta}{T_{\text{env}}}}}\left(\int_{0}^{\delta} dx w(x)\right)\left(\int_{0}^{\delta} dx \frac{1}{w(x)}\right) + \int_{0}^{\delta} dy \frac{1}{w(y)} \int_{0}^{y} w(x) dx\right] + O_{\ell}(1). \quad (120)$$

### B.5.3 The ratio $|\log p_-|/\langle T_X \rangle$

It follows from the asymptotic relations for $\langle T_X \rangle$ and $|\log p_-|$, given by Eqs. (120) and (114), respectively, that the ratio

$$\frac{|\log p_-|}{\langle T_X \rangle} = \frac{f\delta}{\gamma} \frac{1 - e^{\frac{-f\delta}{T_{\text{env}}}}}{\int_{0}^{\delta} dy \, w(y)\left(\int_{y}^{y+\delta} dx' \frac{1}{w(x')}\right)} + O(1/\ell). \quad (121)$$

Using Eqs. (63) and (64) for $\dot{s}$ and $j_{ss}$, respectively, together with the identities

$$\int_0^\delta dy\,\frac{1}{w(y)}\int_0^y dx\,w(x) = \int_0^\delta dy\,w(y)\int_y^\delta \frac{1}{w(x)}dx \tag{122}$$

and

$$e^{-\frac{f\delta}{\mathsf{T}_{\text{env}}}}\int_0^\delta dx\,w(x)\int_0^y dx\,\frac{1}{w(x)} = \int_0^\delta dx\,w(x)\int_\delta^{y+\delta} dx\,\frac{1}{w(x)}, \tag{123}$$

we readily obtain Eq. (67), which is what we were meant to show.

## B.6 Van't Hoff-Arrhenius law near equilibrium

We show that Eq. (67) yields the Van't Hoff-Arrhenius law Eq. (72).

Indeed, if $\ell$ is large enough, then Eq. (67) together with Eq. (114) yields

$$\langle T_X \rangle = \frac{f\ell}{\mathsf{T}_{\text{env}}}\frac{1}{\dot{s}} + O\left(\frac{1}{\ell}\right), \tag{124}$$

where the mean entropy production rate $\dot{s}$ is given by Eq. (63). Since the mean entropy production rate is proportional to the stationary current, given by Eq. (64), we can use saddle point integrals to evaluate the mean current in the limit $\mathsf{T}_{\text{env}} \to 0$ and to obtain the Van't Hoff-Arrhenius law.

Let us therefore first revisit the saddle point method in Sec. B.6.1, and then apply it to the mean current to obtain the Van't Hoff-Arrhenius law in Sec. B.6.2.

### B.6.1 Saddle point integrals in the limit of $\mathsf{T}_{\text{env}} \to 0$

We first revisit briefly the saddle point method.

Let $v(x)$ be a function defined on the interval $[0, \delta]$. We consider integrals of the form

$$\int_0^\delta dx\,e^{\frac{v(x)}{\mathsf{T}_{\text{env}}}}f(x), \tag{125}$$

in the limiting case of small $\mathsf{T}_{\text{env}}$. In this limiting case,

$$\int_0^\delta dx\,e^{\frac{v(x)}{\mathsf{T}_{\text{env}}}}f(x) = \kappa f(x_{\text{max}})e^{\frac{v_{\text{max}}}{\mathsf{T}_{\text{env}}}} + O\left(\frac{\mathsf{T}_{\text{env}}}{v_{\text{max}}}\right), \tag{126}$$

where the prefactor $\kappa$ depends on the properties of the function $v$ at the maximum. Below, we consider four relevant cases for $\kappa$. Note that we use the following notation: if $x_{\text{max}} = \arg\max v(x)$, then $v_{\text{max}} = v(x_{\text{max}})$, $v'_{\text{max}} = v'(x_{\text{max}})$, and $v''_{\text{max}} = v''(x_{\text{max}})$. The four cases are the following:

- $v'_{\text{max}} = 0$ and $x_{\text{max}} \in (0, \delta)$:

$$\kappa = \sqrt{\frac{2\pi \mathsf{T}_{\text{env}}}{-v''_{\text{max}}}}; \tag{127}$$

- $v'_{\text{max}}$ does not exist (maximum is a cusp) and $x_{\text{max}} \in (0, \delta)$:

$$\kappa = \mathsf{T}_{\text{env}}\left(\frac{1}{v^+_{\text{max}}} - \frac{1}{v^-_{\text{max}}}\right), \tag{128}$$

where

$$v^+_{\text{max}} = \lim_{\epsilon \to 0}\frac{v(x_{\text{max}}) - v(x_{\text{max}} - \epsilon)}{\epsilon}, \quad \text{and} \quad v^-_{\text{max}} = \lim_{\epsilon \to 0}\frac{v(x_{\text{max}} + \epsilon) - v(x_{\text{max}})}{\epsilon}; \tag{129}$$

- $x_{max} = 0$:

$$\kappa = -\frac{\mathsf{T}_{env}}{v_{max}^-} ; \tag{130}$$

- $x_{max} = \delta$:

$$\kappa = \frac{\mathsf{T}_{env}}{v_{max}^+} . \tag{131}$$

### B.6.2 The mean first-passage time in the low temperature limit and the linear response limit

To derive the Arrhenius law, we take two limits, viz., the near equilibrium limit $f\delta/\mathsf{T}_{env} \approx 0$ and the low temperature limit $\mathsf{T}_{env} \approx 0$. The order of the limits is such that we take first the near equilibrium limit and then the low temperature limit.

Taking the linear response limit with $f\delta/\mathsf{T}_{env} \approx 0$, we obtain

$$w(x) = e^{-\frac{u(x)}{\mathsf{T}_{env}}} \left( 1 + \frac{f x}{\mathsf{T}_{env}} + O\left( \left( \frac{f\delta}{\mathsf{T}_{env}} \right)^2 \right) \right), \tag{132}$$

and

$$\frac{1}{w(x)} = e^{\frac{u(x)}{\mathsf{T}_{env}}} \left( 1 - \frac{f x}{\mathsf{T}_{env}} + O\left( \left( \frac{f\delta}{\mathsf{T}_{env}} \right)^2 \right) \right), \tag{133}$$

such that

$$j_{ss} = \frac{f\delta}{\gamma} \frac{1}{\int_0^\delta \mathrm{d}y\, e^{-\frac{u(y)}{\mathsf{T}_{env}}} \int_0^\delta \mathrm{d}x\, e^{\frac{u(x)}{\mathsf{T}_{env}}}} + O\left( \left( \frac{f\delta}{\mathsf{T}_{env}} \right)^2 \right). \tag{134}$$

Second, we take the low temperature limit with $\mathsf{T}_{env} \approx 0$. Using the saddle point method, we obtain that

$$j_{ss} = \frac{f\delta}{\gamma} \kappa_1 \kappa_2 e^{-\frac{E_b}{\mathsf{T}_{env}}} + O\left( \left( \frac{f\delta}{\mathsf{T}_{env}} \right)^2 \right) \tag{135}$$

where $\kappa_1$ and $\kappa_2$ are two prefactors due to the two saddle point integrals in Eq. (134). The entropy production rate follows from Eq. (63) and is given by

$$\dot{s} = \frac{(f\delta)^2}{\gamma \mathsf{T}_{env}} \kappa_1 \kappa_2 e^{-\frac{E_b}{\mathsf{T}_{env}}} + O\left( \left( \frac{f\delta}{\mathsf{T}_{env}} \right)^3 \right). \tag{136}$$

Lastly, using Eq. (124) we obtain the Van't Hoff-Arrhenius law for the mean-first passage time

$$\langle T_X \rangle = \frac{\ell}{\delta} \frac{\gamma}{f\delta} \frac{1}{\kappa_1 \kappa_2} e^{\frac{E_b}{\mathsf{T}_{env}}} \left( 1 + O\left( \frac{f\delta}{\mathsf{T}_{env}} \right) \right). \tag{137}$$

We discuss two relevant cases:

- $u'_{max} = u'_{min} = 0$ and $x_{max}, x_{min} \in (0, \delta)$:

$$\kappa_1 \kappa_2 = \frac{\sqrt{-u''_{min} u''_{max}}}{2\pi \mathsf{T}_{env}} ; \tag{138}$$

- $u'_{max} \neq 0$ and $u'_{min} \neq 0$:

$$\kappa_1 \kappa_2 = \left( \frac{1}{u_{max}^+} - \frac{1}{u_{max}^-} \right)^{-1} \left( \frac{1}{u_{min}^+} - \frac{1}{u_{min}^-} \right)^{-1} \frac{1}{\mathsf{T}_{env}^2} . \tag{139}$$

## C   Mean first-passage time for an overdamped Brownian particle in a periodic potential that is triangular and in a uniform force field

We derive a number of explicit formulas that have been used to generate the curves in the Figs. 1-4. Similar to the previous appendix, we consider a Brownian motion in a uniform force field $f$ and a periodic potential $u$, for which dynamics of the position variable $X$ is described by the overdamped Langevin Eq. (61). However, in this appendix we specify the potential of the process, viz., we consider the triangular potential given by Eq. (62), which allows us to derive explicit results.

### C.1   Stationary distribution

For the triangular potential Eq. (62), the stationary probability distribution given by Eq. (91) reads [22]

$$p_{ss}(x) = \begin{cases} a_1 + a_2 e^{\frac{x f_+}{T_{env}}} & \text{if} \quad x \in [0, x^*], \\ a_3 + a_4 e^{\frac{x f_-}{T_{env}}} & \text{if} \quad x \in [x^*, \delta], \end{cases} \tag{140}$$

where

$$f_+ = f - \frac{u_0}{x^*}, \quad \text{and} \quad f_- = f + \frac{u_0}{\delta - x^*}, \tag{141}$$

and

$$a_1 = f_+ f_-^2 \frac{e^{\frac{f_- x^*}{T_{env}}} - e^{\frac{f_- \delta + f_+ x^*}{T_{env}}}}{\mathcal{N}}, \tag{142}$$

$$a_2 = f_+ f_-(f_- - f_+) \frac{e^{\frac{f_- \delta}{T_{env}}} - e^{\frac{f_- x^*}{T_{env}}}}{\mathcal{N}}, \tag{143}$$

$$a_3 = f_+^2 f_- \frac{e^{\frac{f_- x^*}{T_{env}}} - e^{\frac{f_- \delta + f_+ x^*}{T_{env}}}}{\mathcal{N}}, \tag{144}$$

$$a_4 = f_+ f_-(f_+ - f_-) \frac{e^{\frac{f_+ x^*}{T_{env}}} - 1}{\mathcal{N}}, \tag{145}$$

and where the normalisation constant

$$\begin{aligned} \mathcal{N} = &T_{env}(f_+ - f_-)^2 \left( e^{\frac{f_+ x^*}{T_{env}}} - 1 \right) \left( e^{\frac{f_- \delta}{T_{env}}} - e^{\frac{f_- x^*}{T_{env}}} \right) \\ &+ f_+ f_-(f_+ \delta - f_+ x^* + f_- x^*) \left( e^{\frac{f_- x^*}{T_{env}}} - e^{\frac{f_- \delta + f_+ x^*}{T_{env}}} \right). \end{aligned} \tag{146}$$

The stationary current is given by the expression

$$j_{ss} = \frac{f_+ a_1}{\gamma} = \frac{f_- a_3}{\gamma}. \tag{147}$$

In Fig. 6, we plot the stationary distribution $p_{ss}$ for various values of the nonequilibrium driving $f\delta/T_{env}$. Observe that the distribution concentrates around the values $x \approx 0$ or $x \approx \delta$, and thus the process resembles a hopping process, as is also visible in Fig. 1.

## C.2 Mean first-passage time

For the case of a triangular potential, we evaluate explicitly the integrals in Eqs. (109), (110), (111), and (112) leading to an explicit expression for the mean first-passage time $\langle T_X \rangle$ in Eq. (107). In particular, we obtain explicit expressions for the following integrals:

$$
\int_0^z \mathrm{d}x\, w(x) =
\begin{cases}
\dfrac{\mathsf{T}_{\mathrm{env}}}{f_+}\left(e^{\frac{f+z}{\mathsf{T}_{\mathrm{env}}}}-1\right) & \text{if} \quad z<x^*, \\[2ex]
\dfrac{\mathsf{T}_{\mathrm{env}}}{f_+}\left(e^{\frac{f_+x^*}{\mathsf{T}_{\mathrm{env}}}}-1\right)+\dfrac{\mathsf{T}_{\mathrm{env}}}{f_-}e^{-\frac{u_0}{\mathsf{T}_{\mathrm{env}}}\frac{\delta}{\delta-x^*}}\left(e^{\frac{f_-z}{\mathsf{T}_{\mathrm{env}}}}-e^{\frac{f_-x^*}{\mathsf{T}_{\mathrm{env}}}}\right) & \text{if} \quad z>x^*,
\end{cases}
\tag{148}
$$

$$
\int_0^z \dfrac{\mathrm{d}x}{w(x)} =
\begin{cases}
\dfrac{\mathsf{T}_{\mathrm{env}}}{f_+}\left(1-e^{\frac{-f+z}{\mathsf{T}_{\mathrm{env}}}}\right) & \text{if} \quad z<x^*, \\[2ex]
\dfrac{\mathsf{T}_{\mathrm{env}}}{f_+}\left(1-e^{\frac{-f_+x^*}{\mathsf{T}_{\mathrm{env}}}}\right)+\dfrac{\mathsf{T}_{\mathrm{env}}}{f_-}e^{\frac{u_0}{\mathsf{T}_{\mathrm{env}}}\frac{\delta}{\delta-x^*}}\left(e^{\frac{-f_-x^*}{\mathsf{T}_{\mathrm{env}}}}-e^{\frac{-f_-z}{\mathsf{T}_{\mathrm{env}}}}\right) & \text{if} \quad z>x^*,
\end{cases}
\tag{149}
$$

and

$$
\int_{\delta-z}^{\delta} \dfrac{\mathrm{d}x}{w(x)} =
\begin{cases}
\dfrac{\mathsf{T}_{\mathrm{env}}}{f_+}\left(e^{\frac{-f+(\delta-z)}{\mathsf{T}_{\mathrm{env}}}}-e^{\frac{-f_+x^*}{\mathsf{T}_{\mathrm{env}}}}\right)+\dfrac{\mathsf{T}_{\mathrm{env}}}{f_-}e^{\frac{u_0}{\mathsf{T}_{\mathrm{env}}}\frac{\delta}{\delta-x^*}}\left(e^{\frac{-f_-x^*}{\mathsf{T}_{\mathrm{env}}}}-e^{\frac{-f_-\delta}{\mathsf{T}_{\mathrm{env}}}}\right) & \text{if} \quad \delta-z<x^*, \\[2ex]
\dfrac{\mathsf{T}_{\mathrm{env}}}{f_-}e^{\frac{u_0}{\mathsf{T}_{\mathrm{env}}}\frac{\delta}{\delta-x^*}}\left(e^{\frac{-f_-(\delta-z)}{\mathsf{T}_{\mathrm{env}}}}-e^{\frac{-f_-\delta}{\mathsf{T}_{\mathrm{env}}}}\right) & \text{if} \quad \delta-z>x^*.
\end{cases}
\tag{150}
$$

In addition, if $z<x^*$, then

$$
\int_0^z \mathrm{d}y\, \frac{1}{w(y)}\int_0^y w(x)\mathrm{d}y = \frac{\mathsf{T}_{\mathrm{env}}}{f_+}z-\left(\frac{\mathsf{T}_{\mathrm{env}}}{f_+}\right)^2\left(1-e^{-\frac{f+z}{\mathsf{T}_{\mathrm{env}}}}\right),
\tag{151}
$$

and if $z>x*$, then

$$
\begin{aligned}
&\int_0^z \mathrm{d}y\, \frac{1}{w(y)}\int_0^y w(x)\mathrm{d}y \\
&= \frac{\mathsf{T}_{\mathrm{env}}}{f_+}x^*+\frac{\mathsf{T}_{\mathrm{env}}}{f_-}(z-x^*)-\left(\frac{\mathsf{T}_{\mathrm{env}}}{f_+}\right)^2\left(1-e^{-\frac{f_+x^*}{\mathsf{T}_{\mathrm{env}}}}\right)-\left(\frac{\mathsf{T}_{\mathrm{env}}}{f_-}\right)^2\left(1-e^{\frac{f_-(x^*-z)}{\mathsf{T}_{\mathrm{env}}}}\right) \\
&\quad +\frac{\mathsf{T}_{\mathrm{env}}}{f_-}e^{\frac{u_0}{\mathsf{T}_{\mathrm{env}}}\frac{\delta}{\delta-x^*}}\left(e^{-\frac{f_-x^*}{\mathsf{T}_{\mathrm{env}}}}-e^{-\frac{f_-z}{\mathsf{T}_{\mathrm{env}}}}\right)\frac{\mathsf{T}_{\mathrm{env}}}{f_+}\left(e^{\frac{f_+x^*}{\mathsf{T}_{\mathrm{env}}}}-1\right).
\end{aligned}
\tag{152}
$$

Lastly, it holds that

$$
\int_0^{\delta} \mathrm{d}y\, \frac{1}{w(y)}\int_y^{\delta} \mathrm{d}x\, w(x) = \int_0^{\delta} \mathrm{d}y\, \frac{1}{w(y)}\int_0^{\delta} \mathrm{d}x\, w(x) - \int_0^{\delta} \mathrm{d}y\, \frac{1}{w(y)}\int_0^y \mathrm{d}x\, w(x)
\tag{153}
$$

and

$$
\begin{aligned}
\int_{\delta-z}^{\delta} \mathrm{d}y\, \frac{1}{w(y)}\int_y^{\delta} \mathrm{d}x\, w(x) &= \int_0^{\delta} \mathrm{d}y\, \frac{1}{w(y)}\int_0^{\delta} \mathrm{d}x\, w(x) - \int_0^{\delta} \mathrm{d}y\, \frac{1}{w(y)}\int_0^y \mathrm{d}x\, w(x) \\
&\quad - \int_0^{\delta-z} \mathrm{d}y\, \frac{1}{w(y)}\int_0^{\delta} \mathrm{d}x\, w(x) + \int_0^{\delta-z} \mathrm{d}y\, \frac{1}{w(y)}\int_0^y \mathrm{d}x\, w(x).
\end{aligned}
\tag{154}
$$

Substituting the above integrals, given by Eqs. (148)-(154), into Eqs. (109), (110), (111), and (112), and consequently using these in Eq. (107) for $\langle T_X \rangle$, we obtain a closed form expression for $\langle T_X \rangle$.

In the Figs. 3 and 4 of the main text we have used this closed form expression of $\langle T_X \rangle$ to plot $\langle T \rangle \dot{s}/|\log p_-|$ as a function of $\ell$ or $\langle T_X \rangle$ as a function of $\mathsf{T}_{\mathrm{env}}$.

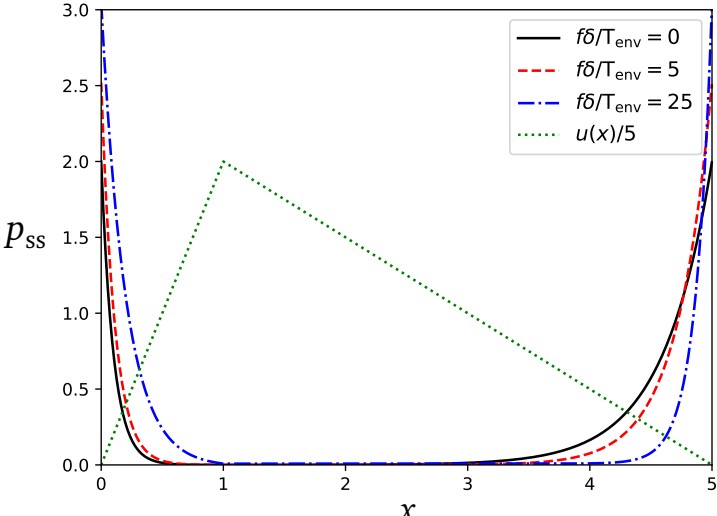

Figure 6: Stationary distribution $p_{\text{ss}}$ for the overdamped Langevin process (61) with triangular potential (62) as a function of $x$ for $\delta = 5$, $x^* = 1$, $u_0 = 10$, $\mathsf{T}_{\text{env}} = 1$ and for given values of $f$. The value of $\gamma$ is immaterial. Solid lines are plots of $p_{\text{ss}}$ obtained from the Eqs. (140)-(146). The green dotted line plots the potential $u$ divided by 5.

### C.3   Recovering the Van't Hoff-Arrhenius law

The Eq. (137) in the particular case of a triangular potential leads to

$$\langle T_X \rangle = \frac{\ell \gamma}{f} \frac{\mathsf{T}_{\text{env}}^2}{u_0^2} e^{\frac{u_0}{\mathsf{T}_{\text{env}}}} \left( 1 + O\left( \frac{f \delta}{\mathsf{T}_{\text{env}}} \right) \right). \tag{155}$$

The green dotted line in Fig. 4 of the main text plots this equation.

## D   Biased hopping process

We determine the splitting probabilities and the moments of the first-passage time $T_X$, defined in Eq. (77), of the biased hopping process $X$, determined by Eq. (73). We make use of the decision variable

$$D = \begin{cases} 1 & \text{if} \quad X(T_X) \geq \ell_+, \\ -1 & \text{if} \quad X(T_X) \leq -\ell_-. \end{cases} \tag{156}$$

### D.1   Martingales in the biased hopping processes

The processes

$$Z(t) = e^{zX(t) + [(1-e^z)k_+ + (1-e^{-z})k_-]t} \tag{157}$$

are martingales for all values of $z \in \mathbb{R}$ (see Appendix A.1 for the definition of a martingale). Indeed, using Itô's formula for jump processes [78], we obtain

$$dZ(t) = (e^z - 1)Z(t)[dN_+(t) - k_+ dt] + (e^{-z} - 1)Z(t)[dN_-(t) - k_- dt], \tag{158}$$

which is a martingale process as both $dN_+(t) - k_+ dt$ and $dN_-(t) - k_- dt$ are martingales. In the special case of $z = \ln \frac{k_-}{k_+}$, we obtain that $Z(t) = e^{-S(t)}$ is the exponentiated negative entropy production, which is an example of martingale process [23].

**Proposition 1** (A martingale equality). *If $k_+ > k_-$, then for all $z \in \mathbb{R} \setminus [\ln \frac{k_-}{k_+}, 0]$ it holds that*

$$1 = \left\langle 1_{T_X < \infty} 1_{D=1} e^{z[\ell_+] + T_X f(z)} + 1_{T_X < \infty} 1_{D=-1} e^{-z[\ell_-] + T_X f(z)} \right\rangle, \tag{159}$$

*where*

$$f(z) = (1 - e^z) k_+ + (1 - e^{-z}) k_-, \tag{160}$$

*and where $[\ell_+]$ and $[\ell_-]$ are the smallest natural numbers that are larger than $\ell_+$ and $\ell_-$, respectively.*

*Proof.* Since $Z(t)$ is a martingale, we can apply Theorem 2 to $Z(t \wedge T_X)$ yielding

$$1 = \langle Z(t \wedge T_X) \rangle = \left\langle e^{zX(t) + (t \wedge T_X) f(z)} \right\rangle. \tag{161}$$

Since for $z \in \mathbb{R} \setminus [\ln \frac{k_-}{k_+}, 0]$ it holds that $f(z) < 0$, the process $Z(t \wedge T_X)$ is bounded from above, viz.,

$$e^{zX(t \wedge T_X) + (t \wedge T_X) f(z)} < e^{z[\ell_+]}. \tag{162}$$

Hence, the bounded convergence theorem applies, see e.g. Ref. [79], and therefore the limit $t \to \infty$ can be taken under the expectation value yielding

$$
\begin{aligned}
1 &= \left\langle \lim_{t \to \infty} e^{zX(t \wedge T_X) + (t \wedge T_X) f(z)} \right\rangle \tag{163} \\
&= \left\langle 1_{T_X < \infty} 1_{D=1} e^{z[\ell_+] + T_X f(z)} + 1_{T_X < \infty} 1_{D=-1} e^{-z[\ell_-] + T_X f(z)} \right\rangle, \tag{164}
\end{aligned}
$$

which completes the proof of the equality (159). □

In what follows, we repeatedly use the martingale equality of Proposition 1 to derive various properties $T_X$.

## D.2 The first-passage time $T_X$ is with probability one finite

**Proposition 2.** *It holds that $T_X$ is almost surely finite, i.e.,*

$$p_- + p_+ = 1. \tag{165}$$

*Proof.* We take the the limit $z \to 0$ in Eq. (159). Since for $z \in [0, 1]$ the argument in the expectation value is bounded by $e^{\ell_+}$, the bounded convergence theorem applies, see e.g. Ref. [79], and

$$
\begin{aligned}
1 &= \lim_{z \to 0} \left\langle 1_{T_X < \infty} 1_{D=1} e^{z[\ell_+] + T_X f(z)} + 1_{T_X < \infty} 1_{D=-1} e^{-z[\ell_-] + T_X f(z)} \right\rangle \\
&= \langle 1_{T_X < \infty} 1_{D=1} + 1_{T_X < \infty} 1_{D=-1} \rangle = \langle 1_{T_X < \infty} \rangle = \mathbb{P}[T_X < \infty],
\end{aligned}
$$

where we have used that $f(0) = 0$. □

## D.3 Splitting probabilities

**Proposition 3.** *It holds that*

$$p_+ = \frac{1 - e^{-[\ell_-] \ln \frac{k_+}{k_-}}}{1 - e^{-([\ell_+] + [\ell_-]) \ln \frac{k_+}{k_-}}}, \quad \text{and} \quad p_- = e^{-[\ell_-] \ln \frac{k_+}{k_-}} \frac{1 - e^{-[\ell_+] \ln \frac{k_+}{k_-}}}{1 - e^{-([\ell_+] + [\ell_-]) \ln \frac{k_+}{k_-}}}, \tag{166}$$

*where $[\ell_-]$ and $[\ell_+]$ are the smallest natural numbers that are greater or equal than $\ell_-$ and $\ell_+$, respectively.*

*Proof.* We apply Theorem 2 to the martingale

$$e^{-S(t)} = e^{X(t)\ln\frac{k_-}{k_+}}, \tag{167}$$

yielding

$$\left\langle e^{X(t\wedge T_X)\ln\frac{k_-}{k_+}} \right\rangle = 1. \tag{168}$$

We can split the quantity $\left\langle e^{X(t\wedge T_X)\ln\frac{k_-}{k_+}} \right\rangle$ into three terms, corresponding to the events $T_X < t$ and $D(T_X) = 1$, $T_X < t$ and $D(T_X) = -1$, and $T_X > t$, yielding in the limit of large $t$,

$$\lim_{t\to\infty}\left\langle e^{X(t\wedge T_X)\ln\frac{k_-}{k_+}} \right\rangle \le p_- e^{-[\ell_-]\ln\frac{k_-}{k_+}} + p_+ e^{[\ell_+]\ln\frac{k_-}{k_+}} + (1-p_--p_+)\langle e^{X(t)\ln\frac{k_-}{k_+}}|T_X > t\rangle. \tag{169}$$

Since the last term is greater or equal than zero, it holds that

$$\lim_{t\to\infty}\left\langle e^{X(t\wedge T_X)\ln\frac{k_-}{k_+}} \right\rangle \ge p_- e^{-[\ell_-]\ln\frac{k_-}{k_+}} + p_+ e^{[\ell_+]\ln\frac{k_-}{k_+}}. \tag{170}$$

Alternatively, we can bound from above the last term of Eq. (169) with the condition $X(t) \ge -\ell_-$ when $T_X > t$, yielding

$$\lim_{t\to\infty}\left\langle e^{X(t\wedge T_X)\ln\frac{k_-}{k_+}} \right\rangle \le p_- e^{-[\ell_-]\ln\frac{k_-}{k_+}} + p_+ e^{[\ell_+]\ln\frac{k_-}{k_+}} + (1-p_--p_+)e^{-[\ell_-]\ln\frac{k_-}{k_+}}. \tag{171}$$

According to Proposition 2, it holds that $p_- + p_+ = 1$, and thus

$$p_- e^{-[\ell_-]\ln\frac{k_-}{k_+}} + p_+ e^{[\ell_+]\ln\frac{k_-}{k_+}} = \lim_{t\to\infty}\left\langle e^{X(t\wedge T_X)\ln\frac{k_-}{k_+}} \right\rangle = 1. \tag{172}$$

The solutions to the Eqs. (165) and (172) are given by Eqs. (166), which completes the proof. □

Using $b = k_-/k_+$ in Eq. (166), we obtain the Eq. (78) in the main text.

## D.4 Generating function

We derive the explicit formula given by Eqs. (80)-(82) for the generating function $g(y)$, as defined in Eq. (79), for $y > 0$.

The generating function $g(y)$ can be written as

$$g(y) = p_+ g_+(y) + p_- g_-(y), \tag{173}$$

where $g_+$ and $g_-$ are the conditional generating functions

$$g_+(y) = \langle e^{-yT_X(k_-+k_+)}|D=1\rangle, \quad\text{and}\quad g_-(y) = \langle e^{-yT_X(k_-+k_+)}|D=-1\rangle. \tag{174}$$

**Lemma 2.** *It holds that*

$$1 = \left(\frac{1}{2}\left[(1+b)(1+y) + \sqrt{-4b+(1+b)^2(1+y)^2}\right]\right)^{[\ell_+]} p_+ g_+(y),$$

$$+ \left(\frac{1}{2}\left[(1+b)(1+y) + \sqrt{-4b+(1+b)^2(1+y)^2}\right]\right)^{-[\ell_-]} p_- g_-(y), \tag{175}$$

*and*

$$1 = \left(\frac{1}{2}\left[(1+b)(1+y) - \sqrt{-4b+(1+b)^2(1+y)^2}\right]\right)^{[\ell_+]} p_+ g_+(y),$$

$$+ \left(\frac{1}{2}\left[(1+b)(1+y) - \sqrt{-4b+(1+b)^2(1+y)^2}\right]\right)^{-[\ell_-]} p_- g_-(y). \tag{176}$$

*Proof.* We rewrite the relation (159) for $z \notin [\ln \frac{k_-}{k_+}, 0]$ as

$$1 = e^{z[\ell_+]} p_+ \langle e^{f(z)T(k_-+k_+)} | D = 1 \rangle + e^{-z[\ell_-]} p_- \langle e^{f(z)T(k_-+k_+)} | D = -1 \rangle. \tag{177}$$

Setting

$$y = -f(z) \tag{178}$$

and solving towards $z$, we obtain two solutions.

First, let us consider the solution branch for $z \geq 0$, which is given by

$$z = \ln \left( \frac{1}{2} \left[ (1+b)(1+y) + \sqrt{-4b + (1+b)^2(1+y)^2} \right] \right). \tag{179}$$

Using Eqs. (178) and (179) in (177), we obtain Eq. (177).

Second, let us consider the solution branch for $z \leq \ln b$, namely,

$$z = \ln \left( \frac{1}{2} \left[ (1+b)(1+y) - \sqrt{-4b + (1+b)^2(1+y)^2} \right] \right). \tag{180}$$

In this case, using Eqs. (178) and (180) in (177), we obtain the Eq. (176).

$\square$

**Proposition 4.** *The generating function Eq. (79) is given by Eqs. (80)-(82).*

*Proof.* We find Eq. (80) readily by solving the Eqs. (176)-(177). $\square$

### D.5 Moments of the first-passage times $T_X$

The moments of first passage times follow from taking explicitly the derivatives in Eq. (83).

The first moment is given by

$$\langle T_X \rangle = \frac{[\ell_+] p_+ - [\ell_-] p_-}{k_+ - k_-}. \tag{181}$$

The second moment is given by

$$
\begin{aligned}
(k_+ - k_-)^2 \langle T_X^2 \rangle &= \frac{p_+}{1 - b^{[\ell_-]+[\ell_+]}} \left( [\ell_+]^2 + [\ell_+] \tanh^{-1}\left( \frac{a}{2T_{\text{env}}} \right) \right) - [\ell_-]^2 p_- \left( \frac{3 + b^{[\ell_-]+[\ell_+]}}{1 - b^{[\ell_-]+[\ell_+]}} \right) \\
&\quad + \frac{p_+ b^{[\ell_-]+[\ell_+]}}{1 - b^{[\ell_-]+[\ell_+]}} \left( 3[\ell_+]^2 - [\ell_+] \tanh^{-1}\left( \frac{a}{2T_{\text{env}}} \right) \right) \\
&\quad + [\ell_-] \tanh^{-1}\left( \frac{a}{2T_{\text{env}}} \right) \frac{b^{2[\ell_-]+[\ell_+]}(1 - b^{[\ell_+]})}{(1 - b^{[\ell_-]+[\ell_+]})^2} - 4[\ell_+][\ell_-] \frac{b^{2[\ell_-]+[\ell_+]}}{(1 - b^{[\ell_-]+[\ell_+]})^2} \\
&\quad + \left( [\ell_-] \tanh^{-1}\left( \frac{a}{2T_{\text{env}}} \right) + 8[\ell_-][\ell_+] \right) \frac{b^{[\ell_-]} b^{[\ell_+]}}{(1 - b^{[\ell_-]} b^{[\ell_+]})^2} \\
&\quad - [\ell_-] \left( \tanh^{-1}\left( \frac{a}{2T_{\text{env}}} \right) + 4[\ell_+] \right) \frac{b^{[\ell_-]}}{(1 - b^{[\ell_-]} b^{[\ell_+]})^2},
\end{aligned}
\tag{182}
$$

where we have used the notation $\tanh^{-1}\left( \frac{a}{2T_{\text{env}}} \right) = 1/\tanh\left( \frac{a}{2T_{\text{env}}} \right)$.

We avoid writing down the expression for $\langle T_X^3 \rangle$, as it is even lengthier than $\langle T_X^2 \rangle$.

## D.6 Moments of the first-passage times $T_X$ in the case of symmetric thresholds

We derive the formulae used to plot the lines in the Fig. 5 of the main text.

In the specific case where $\ell_+ = \ell_- = \ell$, we obtain the simpler expression

$$g(y) = \frac{2^{[\ell]} + 2^{-[\ell]}\left(\beta(y) - \sqrt{-4\frac{k_-}{k_+} + \beta^2(y)}\right)^{[\ell]}\left(\beta(y) + \sqrt{-4\frac{k_-}{k_+} + \beta^2(y)}\right)^{[\ell]}}{\left(\beta(y) - \sqrt{-4\frac{k_-}{k_+} + \beta^2(y)}\right)^{[\ell]} + \left(\beta(y) + \sqrt{-4\frac{k_-}{k_+} + \beta^2(y)}\right)^{[\ell]}}, \qquad (183)$$

for the generating function, where $\beta(y)$ is defined in Eq. (82).

In this case, the mean first-passage time is given by

$$\langle T_X \rangle = \frac{[\ell]}{k_+ - k_-}\frac{1 - b^{[\ell]}}{1 + b^{[\ell]}}, \qquad (184)$$

its second moment

$$\langle T_X^2 \rangle = [\ell]\frac{[\ell] + \frac{k_+ + k_-}{k_+ - k_-} - 6[\ell]b^{[\ell]} + b^{2[\ell]}\left([\ell] - \frac{k_+ + k_-}{k_+ - k_-}\right)}{(k_+ - k_-)^2\left(1 + b^{[\ell]}\right)^2}, \qquad (185)$$

and its third moment

$$\begin{aligned}
\langle T_X^3 \rangle &= \frac{[\ell]}{k_+^3(1-b)^5(1+b^{[\ell]})^3}\Big\{2 + 8b + 2b^2 + 3[\ell](1-b^2) + [\ell]^2(1-b)^2 \\
&\quad + b^{[\ell]}(2 + 2b(4+b) + 15(-1+b^2)[\ell] - 23(-1+b)^2[\ell]^2) \\
&\quad + b^{2[\ell]}(-2 - 2b(4+b) + 15(-1+b^2)[\ell] + 23(-1+b)^2[\ell]^2) \\
&\quad + b^{3[\ell]}(2 + 2b(4+b) + 3(-1+b^2)[\ell] + (-1+b)^2[\ell]^2)\Big\}. \qquad (186)
\end{aligned}$$

Formulae (184)-(186) are plotted in Fig. 5 of the main text.

One readily verifies the thermodynamic uncertainty relation

$$\lim_{[\ell]\to\infty}\frac{\langle T_X^2 \rangle - \langle T_X \rangle^2}{\langle T_X \rangle} = \frac{k_+ + k_-}{(k_+ - k_-)^2} \geq \frac{2}{(k_+ - k_-)\log\frac{k_+}{k_-}}, \qquad (187)$$

where we used the fact that $\log(x) \geq \frac{x-1}{x} \geq \frac{x-1}{x+1}$ with $x = k_+/k_-$.

## D.7 Asymptotic expressions for large thresholds

We determine the splitting probabilities and the first two moments of $T_X$ in the limit $\ell_+, \ell_- \gg 1$ with the ratio $\ell_+/\ell_-$ fixed to a constant value. In particular, we derive the Eq. (84) and the Eq. (85) for the specific cases of $n = 1$ and $n = 2$.

From Eqs. (78), we obtain for the splitting probabilities that

$$p_- = b^{[\ell_-]} + O(b^{[\ell_+]+[\ell_-]}), \quad \text{and} \quad p_+ = 1 + O(b^{[\ell_-]}). \qquad (188)$$

Equation (181) implies that the mean first-passage time

$$\langle T_X \rangle = \frac{[\ell_+]}{k_+ - k_-}\left(1 + O(b^{[\ell_-]})\right), \qquad (189)$$

and from Eq. (182) it follows that the second moment

$$\langle T_X^2 \rangle = \frac{[\ell_+]^2}{(k_+ - k_-)^2}\left(1 + \frac{1}{[\ell_+]\tanh\left(\frac{a}{2T_{\text{env}}}\right)} + O(b^{[\ell_-]})\right). \qquad (190)$$

The Eqs. (188) and (189) imply that

$$\frac{[\ell_+]}{[\ell_-]}\frac{|\log p_-|}{\langle T_X\rangle} = \frac{a}{\mathsf{T}_{\text{env}}}\frac{1}{k_+ - k_-}(1 + O(b^{[\ell_-]})).$$

(191)

We recognise in the above formula the entropy production rate $\dot{s}$ given by Eq. (76), and thus

$$\frac{[\ell_+]}{[\ell_-]}\frac{|\log p_-|}{\langle T_X\rangle} = \dot{s} + O(b^{[\ell_-]}).$$

(192)

Analogously, Eqs. (188) and (190) imply that

$$\frac{[\ell_+]}{[\ell_-]}\frac{|\log p_-|}{\sqrt{\langle T_X^2\rangle}} = \dot{s} + O\left(\frac{1}{[\ell_+]}\right).$$

(193)

The thermodynamic uncertainty relation is governed by the subleading $O(1/[\ell_+])$ term in Eq. (193). Using Eqs. (188) and (190), we obtain the Eq. (86) in the main text. Since,

$$\frac{1}{\tanh(x/2)} \geq \frac{2}{x}$$

(194)

the thermodynamic uncertainty relation [16]

$$\frac{2\langle T_X\rangle}{\langle T_X^2\rangle - \langle T_X\rangle^2} \geq \dot{s}$$

(195)

holds.

In order to find asymptotic expressions for the higher order moments, we analyze in the next subsection the probability distribution of $T_X$ in the limit of large thresholds $\ell_-$ and $\ell_+$.

## D.8 Probability distribution in the asymptotic limit $\ell_\pm \to \infty$

In the present appendix, we derive the asymptotic formula (85) for the moments of $T_X$. In order to derive asymptotic expressions for the moments $\langle T_X^n\rangle$ with $n > 2$, we determine the probability distribution in this limit.

Using that $\zeta_- < \zeta_+$, where $\zeta_\pm$ are defined in Eq. (81), we obtain in the limit $\ell_{\min} \to \infty$ for the generating function given by Eq. (80) the formula

$$g(y) = \left(\frac{2}{\zeta_+(y)}\right)^{[\ell_+]}\left(1 + O\left(\left(\frac{\zeta_-(y)}{\zeta_+(y)}\right)^{[\ell_-]}\right)\right) + \left(\frac{\zeta_-(y)}{2}\right)^{[\ell_-]}\left(1 + O\left(\left(\frac{\zeta_-(y)}{\zeta_+(y)}\right)^{[\ell_-]}\right)\right),$$

(196)

which can be further simplified into

$$g(y) = \left(\frac{2}{\zeta_+(y)}\right)^{[\ell_+]} + O(b^{[\ell_-]}).$$

(197)

Considering that $T$ will be large when both $[\ell_+]$ and $[\ell_-]$ are large, we use that $y \sim \frac{1}{[\ell_{\min}]}$. Therefore,

$$\zeta_+(y) = 2 + 2\frac{1+b}{1-b}y + O(y^2).$$

(198)

Taking the inverse Laplace transform, we obtain up to leading order

$$p_{T_X}(t) = \frac{((k_+ + k_-)t)^{[\ell_+]-1}}{\Gamma([\ell_+])}\left(\frac{1-b}{1+b}\right)^{[\ell_+]}e^{-t(k_+ + k_-)\frac{1-b}{1+b}} + O(b^{[\ell_-]}),$$

(199)

which is the Gamma distribution with shape parameter $[\ell_+]$ and rate $(1-b)/(1+b)$.

If we introduce a new variable,

$$\tau = \frac{(k_+ + k_-)t}{[\ell_+]}, \tag{200}$$

then we get

$$p_{\frac{(k_-+k_+)T_X}{[\ell_+]}}(\tau) \sim \exp\left(-[\ell_+]I(\tau) + O_{[\ell_+]}(1)\right) + O(b^{[\ell_-]}), \tag{201}$$

with the large deviation function

$$I(\tau) = \frac{1-b}{1+b}\tau - \log(\tau) - \log\frac{1-b}{1+b} - 1. \tag{202}$$

The minimum of $I$ is reached when

$$\tau^* = \frac{1+b}{1-b}, \tag{203}$$

in which case $I(\tau^*) = 0$. Expanding $I(\tau)$ around $\tau^*$ we obtain

$$I(\tau) = \frac{\left(\tau - \frac{1+b}{1-b}\right)^2}{2\left(\frac{1+b}{1-b}\right)^2} + O(\tau^3). \tag{204}$$

Hence, the distribution of $p_{T_X}$ is

$$p_{\frac{(k_++k_-)T_X}{[\ell_+]}}(\tau) = \sqrt{\frac{[\ell_+]}{2\pi(\tau^*)^2}} \exp\left(-[\ell_+]\frac{(\tau-\tau^*)^2}{2(\tau^*)^2} + O(\tau^2)\right) + O(b^{[\ell_-]}). \tag{205}$$

For large $[\ell_+]$, the distribution $p_{\frac{(k_++k_-)T_X}{[\ell_+]}}(\tau)$ is centered around $\tau = \tau^*$, and therefore $\frac{(k_++k_-)T_X}{[\ell_+]}$ is a deterministic variable in this limit. The moments of $T$ are thus given by

$$\langle T_X^n \rangle = [\ell_+]^n \frac{(\tau^*)^n}{(k_+ + k_-)^n} + O([\ell_+]^{n-1}) = \frac{[\ell_+]^n}{(k_+ - k_-)^n} + O([\ell_+]^{n-1}). \tag{206}$$

Using the formula for $p_-$, given by Eq. (188), and the expression for $\dot{s}$ in Eq. (76), we recover

$$\frac{[\ell_+]}{[\ell_-]}\frac{|\log p_-|}{\left(\langle T_X^n \rangle\right)^{1/n}} = \dot{s} + O\left(\frac{1}{[\ell_+]}\right), \tag{207}$$

which is also the formula (85) that we were meant to derive.

Note that obtaining an explicit expression for the $1/[\ell_+]$ correction terms in Eq. (85) is more complicated as we need to determine the subleading order terms in Eq. (204), which depend on $b$ and are process dependent. Hence, the moments $\langle T_X^n \rangle$ converge for large thresholds to the universal limit given by Eq. (207) since they are governed by the leading order term in the asymptotic behaviour of $T_X$. On the other hand, the Fano factor

$$\frac{\langle T_X^2 \rangle - \langle T_X \rangle^2}{\langle T_X \rangle} \tag{208}$$

characterising uncertainty depends on the subleading terms and will therefore not converge to a universal limit when the thresholds diverge. This clarifies on an example why the first-passage time relations in the present paper are equalities for $J = S$, while this is not the case for thermodynamic uncertainty relations.

# E   Splitting probabilities of currents $J$ in Markov jump processes

We derive the equality (48) for the splitting probabilities of currents $J$ in Markov jump processes $X$ taking values in a finite set $\mathcal{X}$ in the limit when the thresholds $-\ell_-$ and $\ell_+$ are large. Equation (48) contains the splitting probability

$$p_- = \mathbb{P}[J(T_J(-\ell_-, \ell_+)) < -\ell_-] \tag{209}$$

to hit the negative boundary first, and the splitting probability

$$p_+^\dagger = (\mathbb{P} \circ \Theta)[J(T_J(-\ell_-, \ell_+)) > \ell_+] \tag{210}$$

to hit the positive boundary first in the time-reversed dynamics. We have written explicitly the dependencies on the thresholds in $T_J(-\ell_-, \ell_+)$, as this will be relevant when considering time-reversal arguments. In particular, since the map $\Theta$ changes the sign of $J$, it holds that

$$p_+^\dagger = \mathbb{P}[J(T_J(-\ell_+, \ell_-)) < -\ell_+]. \tag{211}$$

## E.1   Splitting probabilities from a random walk process on the real line

To determine the splitting probabilities $p_-$ and $p_+^\dagger$, we define the discrete-time process

$$J_n = J(n\Delta t), \tag{212}$$

with $n \in \mathbb{N}$. When $\Delta t$ is large enough, then the increments $J_n - J_{n-1}$ are independent and identically distributed random variables drawn from a probability distribution $p_{\Delta_J}(\Delta_j)$, and for Markov jump processes the distribution $p_{\Delta_J}$ is supported on a discrete subset of the real line. Hence, for large threshold values, the splitting probability $p_-$ of $J$ is equal to the splitting probability of the following random walk process defined on the real line,

$$dK = \sum_{j \in \mathbb{Z}} \Delta_j dN_j, \tag{213}$$

where $dN_j$ is a Poisson process with rate $k_j = p_{\Delta_J}(j)$, where $\Delta_j = -\Delta_{-j}$, and $\Delta_0 = 0$. We can thus write

$$p_- = \mathbb{P}[K(T_K(-\ell_-, \ell_+)) < -\ell_-]. \tag{214}$$

Since, by definition, the current $J$ changes sign under time reversal, the time-reversed process is

$$dK^\dagger = -\sum_{j \in \mathbb{Z}} \Delta_j dN_j, \tag{215}$$

and

$$p_+^\dagger = \mathbb{P}[K(T_K(-\ell_+, \ell_-)) < -\ell_+]. \tag{216}$$

In the remaining part of this appendix, we determine, using an approach similar to the one presented in Appendix D, the splitting probabilities $p_-$ and $p_+^\dagger$ of the first-passage time $T_K$ in the random walk process $K$, which for large threshold values $-\ell_-$ and $\ell_+$ are identical to those of $J$. Consequently, we use the obtained expressions for the splitting probabilities to demonstrate that the equality (48) is valid in the limit of large thresholds.

In the calculations we repeatedly use the decision variable

$$D_K = \text{sign}(K(T_K) - K(0)). \tag{217}$$

.

## E.2   Martingales related to $K$

The processes

$$Z(t) = e^{zK(t)+tf(z)}, \tag{218}$$

with

$$f(z) = \sum_{j\in\mathbb{Z}} (1 - e^{z\Delta_j}) k_j \tag{219}$$

are martingales for all values of $z \in \mathbb{R}$. Indeed, applying Itô's formula for jump processes [78] to the latter equation, we obtain

$$\mathrm{d}Z(t) = \sum_{j\in\mathbb{Z}} (e^{z\Delta_j} - 1)Z(t)\big[\mathrm{d}N_j(t) - k_j\mathrm{d}t\big], \tag{220}$$

which is a martingale process as the $\mathrm{d}N_j(t) - k_j\mathrm{d}t$ are martingales.

We also define the time-reversed processes

$$\mathrm{d}Z^\dagger(t) = \sum_{j\in\mathbb{Z}} (e^{-z\Delta_j} - 1)Z(t)\big[\mathrm{d}N_j(t) - k_j\mathrm{d}t\big] \tag{221}$$

that run backwards ($\Delta_j \to -\Delta_j$). Note that the time-reversed process $Z^\dagger$ is related to $Z$ by $z \leftrightarrow -z$.

**Proposition 5** (A martingale equality)**.** *For all values $z \in \mathbb{R}$ for which $f(z) < 0$,*

$$1 = \Big\langle 1_{T_K<\infty} 1_{D_K=1} e^{z\ell_+(1+o_{\ell_{\min}}(1))+T_K f(z)} + 1_{T_K<\infty} 1_{D_K=-1} e^{-z\ell_-(1+o_{\ell_{\min}}(1))+T_K f(z)} \Big\rangle. \tag{222}$$

*Proof.* Since $Z(t)$ is a martingale, we can apply Theorem 2 to $Z(t \wedge T_K)$ yielding

$$1 = \langle Z(t \wedge T_K)\rangle = \Big\langle e^{zK(t\wedge T_K)+(t\wedge T_K)f(z)} \Big\rangle. \tag{223}$$

Since $f(z) < 0$,

$$e^{zK(t\wedge T_K)+(t\wedge T_K)f(z)} < e^{z\ell_+(1+o_{\ell_{\min}}(1))}. \tag{224}$$

Hence, the bounded convergence theorem applies, see e.g. Ref. [79], and we can take the limit $t \to \infty$ under the expectation value to obtain

$$
\begin{aligned}
1 &= \Big\langle \lim_{t\to\infty} e^{zK(t\wedge T_K)+(t\wedge T_K)f(z)}\Big\rangle \tag{225}\\
&= \Big\langle 1_{T_K<\infty} 1_{D_K=1} e^{zK(T_K)+T_K f(z)} + 1_{T_K<\infty} 1_{D_K=-1} e^{zK(T_K)+T_K f(z)}\Big\rangle. \tag{226}
\end{aligned}
$$

For large threshold values $-\ell_-$ and $\ell_+$, we have that

$$K(T_K) = -\ell_-(1 + o_{\ell_{\min}}(1)) \quad \text{if} \quad D_K = -1 \tag{227}$$

and

$$K(T_K) = \ell_+(1 + o_{\ell_{\min}}(1)) \quad \text{if} \quad D_K = 1. \tag{228}$$

Substitution of Eqs. (227) and (228) in Eq. (226) gives readily the equality (222), which completes the proof. $\qquad\square$

In what follows, we use the martingale equality (222) to determine the splitting probabilities $p_-$ and $p_+^\dagger$ of $T_J$.

## E.3 The first-passage time $T_K$ is with probability one finite

**Proposition 6.** *It holds that $T_K$ is almost surely finite, i.e.,*

$$p_- + p_+ = 1 \,. \tag{229}$$

*Proof.* We take the the limit $z \to 0$ in Eq. (222). Since for $z \in [0,1]$ the argument in the expectation value is bounded from above by $e^{\ell_+(1+o_{\ell_{\min}}(1))}$, the bounded convergence theorem applies, see e.g. Ref. [79], and

$$
\begin{aligned}
1 &= \lim_{z \to 0} \left\langle 1_{T_K < \infty} 1_{D_K=1} e^{z\ell_+(1+o_{\ell_{\min}}(1))+f(z)T_K} + 1_{T_K < \infty} 1_{D_K=-1} e^{-z\ell_-(1+o_{\ell_{\min}}(1))+f(z)T_K} \right\rangle \\
&= \left\langle 1_{T_K < \infty} 1_{D_K=1} + 1_{T_K < \infty} 1_{D_K=-1} \right\rangle \\
&= \left\langle 1_{T_K < \infty} \right\rangle = \mathbb{P}(T_K < \infty) \,,
\end{aligned}
$$

where we have used $f(0) = 0$. $\qquad\square$

## E.4 Derivation of the Eq. (48) for the splitting probabilities

**Proposition 7.** *If $z^*$ is a nonzero solution to the equation*

$$f(z^*) = \sum_{j \in \mathbb{Z}} (1 - e^{z^* \Delta_j}) k_j = 0 \,, \tag{230}$$

*then*

$$p_+ = \frac{1 - e^{-\ell_- z^*(1+o_{\ell_{\min}}(1))}}{1 - e^{-(\ell_+ + \ell_-)z^*(1+o_{\ell_{\min}}(1))}} \tag{231}$$

*and*

$$p_- = e^{-\ell_- z^*(1+o_{\ell_{\min}}(1))} \frac{1 - e^{-\ell_+ z^*(1+o_{\ell_{\min}}(1))}}{1 - e^{-(\ell_+ + \ell_-)z^*(1+o_{\ell_{\min}}(1))}} \,. \tag{232}$$

*Proof.* The proof is similar to the one of Proposition 3. The process $e^{K(t)z^*}$ is a martingale as it is of the form Eq. (218). Applying Theorem 2 to the martingale $e^{K(t)z^*}$ yields

$$\left\langle e^{K(t \wedge T_K)z^*} \right\rangle = 1 \,. \tag{233}$$

For large $t$, we obtain the upper bound

$$\lim_{t \to \infty} \left\langle e^{K(t \wedge T_K)z^*} \right\rangle \le p_- e^{-\ell_- z^*(1+o_{\ell_{\min}}(1))} + p_+ e^{\ell_+ z^*(1+o_{\ell_{\min}}(1))} + (1 - p_- - p_+) e^{-\ell_- z^*(1+o_{\ell_{\min}}(1))} \,, \tag{234}$$

where we have made use of the Eqs. (227) and (228) to replace $K$ at the stopping time with either $-\ell_-$ or $\ell_+$, and we have bounded $K(t) \ge -\ell_-$. Similarily, we obtain the lower bound

$$\lim_{t \to \infty} \left\langle e^{X(t \wedge T_X)z^*} \right\rangle \ge p_- e^{-\ell_- z^*(1+o_{\ell_{\min}}(1))} + p_+ e^{\ell_+ z^*(1+o_{\ell_{\min}}(1))} \,. \tag{235}$$

According to Proposition 6, it holds that $p_- + p_+ = 1$, and thus

$$p_- e^{-\ell_- z^*(1+o_{\ell_{\min}}(1))} + p_+ e^{\ell_+ z^*(1+o_{\ell_{\min}}(1))} = 1 \,. \tag{236}$$

The solutions to the Eqs. (229) and (236) are given by Eqs. (231) and (232), which completes the proof. $\qquad\square$

**Proposition 8.** *If $z^*$ is a nonzero solution to the equation*

$$f(z^*) = \sum_{j\in\mathbb{Z}}(1 - e^{z^*\Delta_j})k_j = 0,\tag{237}$$

*then*

$$p_+^\dagger = e^{-\ell_+ z^*(1+o_{\ell_{\min}}(1))}\frac{1 - e^{-\ell_- z^*(1+o_{\ell_{\min}}(1))}}{1 - e^{-(\ell_++\ell_-)z^*(1+o_{\ell_{\min}}(1))}}\tag{238}$$

*and*

$$p_-^\dagger = \frac{1 - e^{-\ell_+ z^*(1+o_{\ell_{\min}}(1))}}{1 - e^{-(\ell_++\ell_-)z^*(1+o_{\ell_{\min}}(1))}}\,.\tag{239}$$

*Proof.* Applying the Proposition 7 to the Eq. (216), and using the fact that $z^*$ is independent of the threshold values $\ell_-$ and $\ell_+$, we readily obtain the equalities (238) and (239). □

**Proposition 9.** *The splitting probabilities $p_-$ and $p_+^\dagger$ obey the equality Eq. (48).*

*Proof.* Eq. (48) follows readily from Eqs. (232) and (238).

□

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
