# Peer review of "Universal tradeoff relation between speed, uncertainty, and dissipation in nonequilibrium stationary states"

_SciPost Physics, doi:SciPost Phys. 12, 139 (2022)_

## Round 1 · Referee Report · Anonymous (Referee 1) · 2021-4-22

Strengths

1) This work is a timely and interesting contribution to the study of thermodynamic trade-offs in stochastic thermodynamics

Weaknesses

1) At times the certain steps of the derivation were a bit unclear

2) The interpretation of the results would benefit from some more precise language

Report

probability that the current first exits that open interval through the lower (or more precisely rarer) value, i.e., the splitting probability. After comparing the predictions of the current work to previous results, the author looks at the results in a model of a Brownian particle in a periodic ratchet. The author also claims that near equilibrium their inequality reduces to the Arrhenius Law for transitions between metastable states.

Overall, the paper is clear and concise, with much technical detail relegated to the appendix. The topic is also timely as there has been a lot of energy in the stochastic thermodynamics community in deriving and analyzing trade-offs between observables and steady state dissipation. This has been driven in large part by the prediction of the thermodynamic uncertainty relation and its modifications. Concurrently, the work of the present author and others exploiting the martingale property of the exponential of the entropy production, has led to a number of novel predictions regarding the fluctuations of entropy production. The current paper goes beyond these works in a number of important ways. By including information like the splitting probability to the study of current first-passage-times, the author presents tighter and more refined bounds than currently in the literature. With that I would like to see the current manuscript published after the author takes some time to tighten some of the language and claims of the paper.

Requested changes

1) In the abstract the author claims the current work is a “time analogue of the thermodynamic uncertainty relations”. It was my understanding that Ref. [47] was a first-passage-time trade-off directly based on the uncertainty relation. In this regard, can one really consider the current inequality a thermodynamic uncertainty relation, or even an analogue, in light of the current terms already coined. The author’s inequality includes information, like a splitting probability, over and above what is included in the thermodynamic uncertainty relation, which is the reason their results are tighter. A more refined description is recommended.

2) In the introduction, the author lists examples of studies of first-passage times in “specific examples of nonequilibrium stochastic processes”. A slight oversight has been the exclusion of the body of work of Mark Dykman and collaborators on escape rates from driven systems, based largely on the eikonal approximation to path integrals.

3) The author introduces the notion of “dissipative current” in this paper. I wonder what is different between a current as usually defined in stochastic thermodynamics and “dissipative currents”. This was especially confusing for me in the first sentence of section 2, because I was not familiar with this terminology.

4) In attempting to provide an interpretation to Eq. (3) in section 2, the author calls the splitting probability p_ (p sub minus), the “reliability”. I hadn’t seen such interpretation before, and I wonder if the author could be more explicit about what is reliable. That is if p_ is smaller in what way is a generic nonequilibrium system more “reliable”. Alternatively, could the author offer some examples where there is an a priori notion of reliability and p_ is a natural measure of it.

5) Following eq. (6), the author writes in mathematical notation that S(t) is in the set O(t). I couldn’t find the definition of this set O(t).
6) In Eq. (7) the author identifies the entropy production as the log of the “ratio between the probabilities densities of the trajectory X_0^t in the forward and backward dynamics”. Here the probability of the backward dynamics is notated by a p with a tilde. In was my understanding that for the stationary, overdamped, Markovian dynamics that the author considers the “forward and backward dynamics” were the same. Instead, to identify the entropy production, one looked at the ratio of the probabilities between the forward and reverse trajectories in the same steady-state ensemble.

7) In eq. (19), I’m a bit confused by the definition of p_. The author uses a large deviation form for the current distribution, which is not normalized. Without this normalization, is p_ still properly defined?

8) The author writes following eq. (20) that the inequality in (20) was derived in three works, Refs. [16,17,43]. Certainly, all three of those works relate to the inequality, but my reading of the literature is that Ref. [17] is the one that derived it while the other two either utilized or conjectured it.

9) I couldn’t see how to arrive at Eq. (23).

10) The author cites Ref. [33] in referring to the martingale property of the exponential of the entropy production. I was under the understanding that the first appearance of this idea was in Chetrite/Gupta J. Stat. Phys. 143, 543 (2011). Same comment in section A.3.

11) In the final sentence of the first full paragraph following eq. (33) the author compares two ratios, calling one “universal” and the other “system-dependent”. What does the author mean here by “universal” and “system-dependent”? The ratio coming from the uncertainty relation is part of an inequality that holds for any nonequilibrium system. Isn’t that universal? The numerical value of the ratios depends specifically on the system parameters. Isn’t that “system-dependent”?

12) The author claims that their inequality reduces to an Arrhenius rate near equilibrium. This is “shown” in section 7. But this is just for a simple model. In what way is this general? If not, what are the assumptions beyond near equilibrium that are required? Are they the same or different from the textbook derivation of the Arrhenius rate?

13) Section 8 deals with inference of entropy production. This is an interesting problem, currently of great interest within the stochastic thermodynamics community. However, I struggle with what the author means here by inference. They study a problem where the observed current is simply proportional to the entropy production. If one can monitor the entropy production like this, there simply is no issue in measuring it directly. Why even consider going through a bound then? The challenging experimental question is when you can’t monitor the entropy production, but instead one can only observe a current that is not simply related to it. Related to this, is in the last sentence in the first paragraph of the discussion, where the author reiterates in the context of entropy inference the inequalities only hold for large enough times. In an experimental situation, how would we know what is “long enough”? Is there a test? Without answering these questions I don’t see how the author can support their claim that their inequality allows entropy production inference.

  • validity: top
  • significance: high
  • originality: high
  • clarity: good
  • formatting: good
  • grammar: excellent

Author:  Izaak Neri  on 2021-08-19  [id 1694]

(in reply to Report 1 on 2021-04-22)
Category:
answer to question

I appreciate the positive comments of the referee and thank the referee for providing constructive comments that give me the opportunity to improve the manuscript.

Below you can find an answer to all the queries raised by the referee:

1) By analogue I meant to say that both the derived first passage bounds and the thermodynamic uncertainty relations are tradeoff relations between speed, uncertainty, and dissipation.

Nevertheless, I do agree with the referee that it may be confusing to speak of uncertaintiy relation analogues especially since these terms have already been coined before. Therefore, I have removed this wording from the abstract and the introduction.

2) I thank the referee for giving me the opportunity to improve the reference list. I have added, among others, the interesting papers https://www.nature.com/articles/45492 and https://journals.aps.org/pre/abstract/10.1103/PhysRevE.73.061109 to the bibliography.

3) Nondissipative currents exist when some variables have odd parity under time reversal. Consider for example an underdamped Brownian particle driven by an external force.

In this case the stochastic process $K = \int^t_0 P\circ dX$, with $P$ the momentum of the particle and $X $ its position, is a nondissipative current: It is a flux as it changes sign when we reverse the order of events while ignoring the parity of the variables; it is not a dissipative current as it does not change sign under time reversal; and its average value is nonzero as $dX$ has on average the same sign as $P$.

Nondissipative currents also appear in the Fokker-Planck equation of the probability density $p(X,P)$. This equation reads as $\partial_t p = -\nabla \dot J$ where $J$ is the average probability flux. For an overdamped system in equilibrium $J=0$. However, in underdamped processes it is possible to have in equilibrium $J$ different than zero and still $\nabla J = 0$. This is due to the existence of nondissipative currents that do not produce entropy.

Since nondissipative currents do not produce entropy, they will not obey the first-passage time trade-off relations derived in the manuscript.

In the new version of the manuscript I elaborate on the distinction between dissipative and nondissipative currents in Appendix A.

4) Consider for example a (biological) cell that is in a state A and let"J" be the net number of proteins of a certain kind produced by the cell. If "J" goes above a certain threshold, then the cell changes its state from A to B. If "J" goes below another threshold, then the cell changes its state from A to C. The splitting probability $p_-$ denotes the reliability of this decision process. If $p_- -> 0$, then we are certain that the cell will change from A to B and the reliability of the process is high. On the other hand, if the noise in the process is large enough so that $p_- \approx 0.5$, then the decision of the cell will be unreliable.

One may think of other examples, for instance, an electronic gate that changes its output state when the current goes above or below a certain threshold, or a molecular motor that copies a part of the genetic information . In the latter case $p_-$ represents the probability of having an error in the process.

5) The symbol O(t) is the Big O notation. It represents a function $f$ for which there exist a constant $M$ so that $|f(t)|< Mt$ when $t$ is large enough. I have clarified the notation in the new version of the manuscript.

6) The derivations in the paper do not require that $X$ is an overdamped process as I have clarified in the new version of the mansucript. The derivation also does not require that the probability distribution in the time-reversed process is the same as the probability distribution in the forward process. A charged particle in a magnetic field is an example of a stationary process for which the time-reversed process is not identical to the forward process as the magnetic field changes sign under time reversal.

7) The normalisation constant can be determined using the saddle point method. It contributes a prefactor, but this prefactor does not contribute to $|log p_-|$ in the limit of large thresholds as it is subleading. I have clarified that in the new version of the manuscript.

8) Yes that is correct. I have rephrased the sentence in order to avoid any misconception: "the large deviation function of the current that holds for overdamped Markov processes in nonequilibrium stationary states, see Refs.[16,17,43]."

9) This is a saddle point calculation. The large deviation function is convex and therefore its minimum value in the interval $(-infty,-ell_-/tau_-]$ is reached at $-ell_-/tau_-$. Using the saddle point method we replace the integral by its integrand evaluated at $-ell_-/tau_- = -dot{s}$. Lastly, we use the Gallavoti-Cohen relation to replace $J(-dot{s})$ by $\dot{s}$. I have clarified derivation in the new version of the manuscript by adding additional steps to the derivation.

10) I agree with the referee. Note that below equation (6), when the martingale property was mentioned first, the paper of Chetrite/Gupta appeared as the first reference. I think later it was not cited because I was building on the integral-fluctuation-relation-at-stopping-times method that was developed in reference [33]. But, in any case, the Chetrite/Gupta paper is a fantastic paper and encouraged by the referee I have taken the opportunity to cite more often in the manuscript.

11) If $ J=S$, then the inequality in the present paper is an equality. As a consequence, if you put all observables of the equality to the left hand side, then the resultant ratio is equal to one and thus universal/system independent. On the other hand, since the thermodynamic uncertainty relation is not tight for J=S , the ratio of the observables that appears in the inequality is dependent on the system parameters (even though the bound is universal). This was illustrated in the figure 4 of the previous manuscript: observe that the ratio converges to one for the first-passage bounds while it converges to a system dependent number for the uncertainty relation. The uncertainty bound itself is universal, but that does not mean that the ratio of the observables appearing in the inequality is.

12) The Arrhenius formula is an empirical formula that describes rate processes in a large number of systems in equilibrium, see P. Hanggi, P. Talkner and M. Borkovec, Reaction-rate theory: fifty years after kramers, Rev. Mod. Phys. 62, 251 (1990), doi:10.1103/RevModPhys.62.251.

The formula has been derived in the paper "H. A. Kramers, Brownian motion in a field of force and the diffusion model of chemical reactions, Physica 7(4), 284 (1940)" for a specific model. It is my understanding that you need a model to derive this formula and the calculations in the Kramers paper are usually onsidered as a demonsration of the Arrhenius formula for equilibrium systems. In the present manuscript, I have considered a nonequilibrium version of the model considered by Kramers.

The Kramers model is one-dimensional, which makes it easy to define the barrier. If you have a multidimensional system then the barrier E_b is path dependent which makes the derivation more complicated. Nevertheless, if there is a clear barrier E_b that seperates two metastable states, then the Arrhenius law should be recovered as the system is effectively one-dimensional.

13) Since the present paper is already long and contains many novel results, I decided to write a separate paper on the use of the derived first-passage time inequalities for the inference of the entropy production rate. In that paper the comments of the referee will be addressed in detail.

The present paper focuses instead on the derivation of novel inequalities and equalities that describe thermodynamic tradeoffs between speed, dissipation and uncertainty.

I have removed all comments in the paper related to thermodynamic inference and I think this resolves the concerns of the referee about unsupported claims.

---

## Round 1 · Referee Report · Anonymous (Referee 2) · 2021-5-11

Report

The author derives bounds on first-passage times of reaching a threshold for currents in a NESS. The author emphasizes that beyond describing a trade-off relation these bounds are useful inference tools for the total entropy production especially in addition to the well-established thermodynamic uncertainty relations. When the system is near equilibrium the author shows that the derived bounds involve the Van’t Hoff-Arrhenius law and hence, he extends this law to a NESS. Furthermore the author discusses the quality of the bounds by deriving equality conditions and by illustrating these bounds for an overdamped Langevin system. The main results are compared with the thermodynamic uncertainty relation and also with several previously derived bounds.

The topic is timely and should be of interest to the broad community interested in the statistical physics of non-equilibrium steady states. There are a few issues, which deserve attention before a final recommendation for publication can be made.

  1. In a NESS eq. (8) is exact without the Landau symbol even for finite times, right?

  2. In eq.(16) the author states a fundamental relation, on which the main results are based. Its validity is motivated by pointing out that the current becomes deterministic. It would help to either quote a references, where this is shown explicitly or to provide a derivation.

  3. A crucial ingredient for getting eq. 3 is the bound eq. 20 on the large deviation function. The very same bound is used for deriving the thermodynamic uncertainty relation. Hence, there seems to be a deep connection between the bounds derived here and the thermodynamic uncertainty relation. When the author points out that eq. 3 can generically be saturated whereas the thermodynamic uncertainty relation cannot, it seems to contradict the fact that the derivation of both bounds requires the same inequality. A further clarification here would be helpful.

  4. The repeated statements concerning the fact that one can infer the entropy production exactly by looking at the first passage time of this current seem to be trivial. If I know that the current I am observing is the entropy current, then I just take its mean and I am done without measuring a first passage time at all. If the author wants to stress that it is sufficient to know that the observed current is proportional to the entropy current, he should clarify this throughout the manuscript. Operationally, in a complex system, the challenge then still is how one would know that the current under investigation is proportional to the entropy current.

  5. The author points out that "an important distinction between the thermodynamic uncertainty relations, Eqs. (31) and Eq. (33), and the bound Eq. (3) on the moments of first-passage times, is that the former are loose bounds while the latter is a tight bound". I guess he means that his results can generically be saturated arbitrary far away from equilibrium whereas the thermodynamic uncertainty relation cannot. When choosing the current as the total entropy production this statement may be true. However, for a current that is not proportional to the entropy production it is not clear a priori how tight the present bounds will be. I strongly recommend to analyze a multi-cyclic system in order to check that for a generic current that is not the entropy current (or proportional to it) the present bound does better than the one based on the thermodynamic uncertainty relation. Without such an analysis, the insinuated superiority seems not to be justified.

  • validity: -
  • significance: -
  • originality: -
  • clarity: -
  • formatting: -
  • grammar: -

Author:  Izaak Neri  on 2021-08-18  [id 1691]

(in reply to Report 2 on 2021-05-11)

**The referee writes:**
>The author derives bounds on first-passage times of reaching a threshold for currents in a NESS. The author emphasizes that beyond describing a trade-off relation these bounds are useful inference tools for the total entropy production especially in addition to the well-established thermodynamic uncertainty relations. When the system is near equilibrium the author shows that the derived bounds involve the Van’t Hoff-Arrhenius law and hence, he extends this law to a NESS. Furthermore the author discusses the quality of the bounds by deriving equality conditions and by illustrating these bounds for an overdamped Langevin system. The main results are compared with the thermodynamic uncertainty relation and also with several previously derived bounds. The topic is timely and should be of interest to the broad >community interested in the statistical physics of non-equilibrium steady states.

**Response:**

I would like to thank the referee for positively commenting on the paper.

**The referee writes:**
>There are a few issues, which deserve attention before a final recommendation for publication can be made.

**Response:**

Thank you for raising issues and giving me the opportunity to improve the paper.

**The referee writes:**
>1. In a NESS eq. (8) is exact without the Landau symbol even for finite times, right?

**Response:**

Agreed. In the new version the Landau symbol has been removed.

**The referee writes:**
>2. In eq.(16) the author states a fundamental relation, on which the main results are based. Its validity is motivated by pointing out that the current becomes deterministic. It would help to either quote a references, where this is shown explicitly or to provide a derivation.

**Response:**

If the current satisfies a large deviation principle, then it is deterministic in the limit of large “t” in the sense that “J/t” has a standard deviation that converges to zero for large enough “t”.

A large deviation principle for current variables in Markov processes, in particular Markov jump processes and diffusion processes that are homogeneous and ergodic, is derived in e.g. A C Barato and R Chetrite, J.Stat Phys 160, 1154-1172 (2015).

I have clarified this in the new version of the paper and added a couple of references.

**The referee writes:**
>3. A crucial ingredient for getting eq. 3 is the bound eq. 20 on the large deviation function. The very same bound is used for deriving the thermodynamic uncertainty relation. Hence, there seems to be a deep connection between the bounds derived here and the thermodynamic uncertainty relation. When the author points out that eq. 3 can generically be saturated whereas the thermodynamic uncertainty relation cannot, it seems to contradict the fact that the derivation of both bounds requires the same inequality. A further clarification here would be helpful.

**Response:**

The thermodynamic uncertainty relation deals with the second moment of the current and thus with the properties of the large deviation function around the origin “z=0”. It is well known that at the origin the bound on the large deviation function is not tight, see for example Pietzonka, Patrick, Andre C. Barato, and Udo Seifert. "Universal bounds on current fluctuations." Physical Review E 93.5 (2016): 05214. On the other hand, if J=S, then the first-passage bound in the present paper evaluates the large deviation function at the point "z = -dot{s}" where the large deviation function bound is tight (see section 5.1 of the previous manuscript). The fact that the large deviation function bound is tight at z=-dot{s} has also been noticed in Ref. Pietzonka, Patrick, Andre C. Barato, and Udo Seifert. "Universal bounds on current fluctuations." Physical Review E 93.5 (2016): 052145, see the lower panel of Figure 3 in that paper which shows the tightness of the bound at xi=-1 in the notation of that paper.

**The referee writes:**
>4. The repeated statements concerning the fact that one can infer the entropy production exactly by looking at the first passage time of this current seem to be trivial. If I know that the current I am observing is the entropy current, then I just take its mean and I am done without measuring a first passage time at all. If the author wants to stressthat it is sufficient to know that the observed current is *proportional* to the entropy current, he should clarify this throughout the manuscript. Operationally, in a complex system, the challenge then still is how one would know that the current under investigation is proportional to the entropy current.

**Response:**

Since for J=S the inequality Eq.(3) is tight, we can infer S by minimising the left hand side of the inequality. Nevertheless, I agree with the referee that the paper does not explore this avenue and therefore the statements about entropy inference remain a bit speculative. For this reason, I have decided to remove the statements about thermodynamic inference and instead I will discuss those in-depth in a separate manuscript.

**The referee writes:**
>5. The author points out that "an important distinction between the thermodynamic uncertainty relations, Eqs. (31) and Eq. (33), and the bound Eq. (3) on the moments of first-passage times, is that the former are loose bounds while the latter is a tight bound". I guess he means that his results can generically be saturated arbitrary far away from equilibrium whereas the thermodynamic uncertainty relationcannot. When choosing the current as the total entropy production thisstatement may be true. However, for a current that is not proportional to the entropy production it is not clear a priori how tight the presentbounds will be. I strongly recommend to analyze a multi-cyclic system in order to check that for a generic current that is not the entropy current (or proportional to it) the present bound does better than the one based on the thermodynamic uncertainty relation. Without such an analysis, the insinuated superiority seems not to be justified.

**Response:**

I agree with the referee that there is no guarantee that for an arbitrary J the bound is “superior” to the uncertainty relation and I did not intend to imply that. The paper shows that for J=S the bound is superior in the sense that it is an inequality and that is all.

I have remedied the concerns of the referees by rephrasing sentences: the new version of the manuscript clearly states that tightness only holds when J=S and any sentence that may have insinuated superiority outside this case has been removed.

It would certainly be very interesting to study a multicyclic system. I would prefer to discuss this in a separate paper together with the previous points on thermodynamic inference and hope the referee will agree with that.

---

## Round 2 · Referee Report · Anonymous (Referee 2) · 2021-11-17

Report

I am satisfied with the response of the author and
the corresponding changes.

---

## Round 2 · Referee Report · Anonymous (Referee 3) · 2021-12-10

Report

Referee report
I. Neri, "Universal tradeoff relation between speed, uncertainty, and dissipation in nonequilibrium stationary states"

This manuscript concerns the statistics of currents that can be observed in non-equilibrium systems in a steady state. This statistics is then put in context with the overall dissipation of the system. Such considerations are currently a hot topic, with the thermodynamic uncertainty relation (TUR) being the most prominent result. The present work goes beyond the standard TUR: instead of the variance of a current it characterizes the mean first passage time for the current to leave a set interval. The main result is formulated as an inequality, which becomes an equality when the current of interest is proportional to the entropy production. The resulting trade-off relation can, in some contexts, be interpreted as an extension of the Arrhenius law.

I think the manuscript is well written. The exposition of the main results in section 2 gives a useful overview, and the following derivation is thorough. The result opens a new pathway for experimental applications and further theoretical work, focusing on the interplay between first passage fluctuations and the energetics of nonequilibrium systems. Moreover, the tools used for the derivation, large deviation theory, martingale theory, and sequential hypothesis testing, complement each other and are of interest in their own right, providing a novel link between different research areas.

The manuscript has already successfully passed two rounds of refereeing, and I it should now be ready for acceptance.

One suggestion: In the introductory section (or the examples of Secs. 8 and 9), it might be helpful to discuss practical issues with the application of the relation. When taking the limit $\ell_-$ to infinity, the probability to leave the interval on the left boundary decreases exponentially. What would determine a good choice for a finite $\ell_-$, given a limited sampling capacity?

In addition, I have a few technical comments:

Below (19), $j_{ss}$ is introduced as "stationary probability flux", but it may not be obvious how this is defined.

For (29), it may be worth reminding the reader that the limit $\ell_-$ to infinity is taken, such that the saddle point approximation ("taking the maximum of the exponent") applies.

Typo "the the" before (40)

Below (40): "which clarifies the notation of Eq. (92)" it is somewhat strange to discuss this here, when the notation appears so much later. Maybe it would be better to refer back to this around (92). Or explain specifically what notation will be used in (92), without referring to the equation yet.

In the statement of Theorem 1, state which values the variable r can take (I assume integers or positive reals)

Last paragraph of p. 15: I think $z\approx 0$ should instead read $z\approx \bar j$ (1st instance) and $z\approx \dot s$ (2nd instance). Also, "or loose"->"are loose"

Fig. 5: In the legend, write "uncertainty relation" (or "TUR") instead of "uncertainty" (otherwise one could think the graph shows the relative uncertainty or similar).
  • validity: -
  • significance: -
  • originality: -
  • clarity: -
  • formatting: -
  • grammar: -

Author:  Izaak Neri  on 2022-02-09  [id 2177]

(in reply to Report 2 on 2021-12-10)

I would like to thank the Referee for commenting positively on the manuscript and providing several useful comments.

Below you can find a point-by-point reply:

*) Referee: "One suggestion: In the introductory section (or the examples of Secs. 8 and 9), it might be helpful to discuss practical issues with the application of the relation. When taking the limit ℓ− to infinity, the probability to leave the interval on the left boundary decreases exponentially. What would determine a good choice for a finite ℓ−, given a limited sampling capacity?"

Reply:
This is indeed an important issue and should be addressed at some point. However, it concerns the practical issue of estimating the quantities in the inequality (3) and not the derivation of this inequality. When writing the paper I have made the decision to focus on theory. The paper derives the main results (3) and (6) and discusses their physical relevance. This is not to say that practical implementation of these inequalities in experiments are not important, but it would require an in-depth study on several examples. At present, the number of examples provided in the paper are too limited to address questions of the form "what is a good choice of $\ell_-$". Nevertheless, I will address this in a future work that focuses on practical questions, and for now, I have added the following sentence in the conclusion:

" In particular, the probability $p_-$ decreases exponentially with $\ell_-$, which raises the question how $p_-$ can be estimated at large values of $\ell_-$."

*) Referee: Below (19), jss is introduced as "stationary probability flux", but it may not be obvious how this is defined.

Reply: The equation (19) has been removed altogether as it is not used.

*) Referee: For (29), it may be worth reminding the reader that the limit ℓ− to infinity is taken, such that the saddle point approximation ("taking the maximum of the exponent") applies.

Reply: Absolutely. This reads now (22) and I have added:

“ For large values of $\ell_-$, the expression Eq.~(22) is a saddle point integral, and hence it is determined by the maximum of the exponent, i.e.,”

*) Referee: Typo "the the" before (40)

Reply: Fixed!

*) Referee: Below (40): "which clarifies the notation of Eq. (92)" it is somewhat strange to discuss this here, when the notation appears so much later. Maybe it would be better to refer back to this around (92). Or explain specifically what notation will be used in (92), without referring to the equation yet.

Answer: This was a typo (overlapping references). Eq.(92) should have read (40), and this has been fixed.

*) Referee: In the statement of Theorem 1, state which values the variable r can take (I assume integers or positive reals)

Reply: Good point. The values of r are integers, and this is now mentioned.

*) Referee: Last paragraph of p. 15: I think z≈0
should instead read z≈¯j (1st instance) and z≈˙s (2nd instance). Also, "or loose"->"are loose"

Reply: The referee is correct. I have fixed this.

*) Referee: Fig. 5: In the legend, write "uncertainty relation" (or "TUR") instead of "uncertainty" (otherwise one could think the graph shows the relative uncertainty or similar).

Reply: Fair point. Has been modified as suggested by the Referee.

Anonymous on 2022-02-10  [id 2182]

(in reply to Izaak Neri on 2022-02-09 [id 2177])

I have no objections to the answers given by the author and the changes made in the manuscript. I therefore agree that it should now be ready for acceptance.

---

## Round 2 · List of Changes

The main changes are the following:

*) The Appendix B from the old manuscript, which was necessary to derive condition (26), has been removed.
Instead, the derivation in the new version of the manuscript relies on the condition (12), which states that J converges asymptotically to a drift-diffusion process, and the first-passage duality of a drift-diffusion process as derived in the new Appendix B.
The new derivation of the inequality is much simpler and more transparent [it relies on the condition (12) that is valid whenever J has finite memory].

*) Section 6 is novel. It provides an alternative derivation of the main results based on resulst from sequential hypothesis testing.

*) The introduction and discussion sections have been simplified, hopefully addressing more clearly the main points of the paper

*) The title has changed. Since the derived results express a tradeoff betweeen speed, uncertainty and dissipation, I thought that the present title better reflects the main point of the paper.

*) Several minor changes addressing the Referee's comments throughout the text.

---

## Round 3 · Referee Report · Anonymous (Referee 3) · 2022-2-22

Report

I have no objections to the answers given by the author and the changes made in the manuscript. I therefore agree that it should now be ready for acceptance.

---

## Round 3 · Author Response

The minor comments of the Referees have been implement, and some steps in the derivations have been further simplified and made more clear. In my opinion, the manuscript is now ready for publication.

---

## Round 3 · List of Changes

The following changes have been implemented:

*) some steps in the derivation of (3) that are based on large deviation theory have been simplified, which leads to a better understanding of (3). In particular, it is shown that (3) follows directly from (19), and there is no need to study the first-passage times at the negative boundary.

*) the derivation of formula (49) has been detailed in Appendix E

---

## Editorial Decision

published